# Appearance of green tea compounds in plasma following acute green tea consumption is modulated by the gut microbiome in mice

John D. Sterrett,[1,2] Kevin D. Quinn,[3] Katrina A. Doenges,[3] Nichole M. Nusbacher,[4] Cassandra L. Levens,[5] Mike L. Armstrong,[3] Richard M. Reisdorph,[3] Harry Smith,[3] Laura M. Saba,[3] Kristine A. Kuhn,[5] Catherine A. Lozupone,[4] Nichole A. Reisdorph[3]

**ABSTRACT**  Studies have suggested that phytochemicals in green tea have systemic anti-inflammatory and neuroprotective effects. However, the mechanisms behind these effects are poorly understood, possibly due to the differential metabolism of phytochemicals resulting from variations in gut microbiome composition. To unravel this complex relationship, our team utilized a novel combined microbiome analysis and metabolomics approach applied to low complexity microbiome (LCM) and human colonized (HU) gnotobiotic mice treated with an acute dose of powdered matcha green tea. A total of 20 LCM mice received 10 distinct human fecal slurries for an $n = 2$ mice per human gut microbiome; 9 LCM mice remained un-colonized with human slurries throughout the experiment. We performed untargeted metabolomics on green tea and plasma to identify green tea compounds that were found in the plasma of LCM and HU mice that had consumed green tea. 16S ribosomal RNA gene sequencing was performed on feces of all mice at study end to assess microbiome composition. We found multiple green tea compounds in plasma associated with microbiome presence and diversity (including acetylagmatine, lactiflorin, and aspartic acid negatively associated with diversity). Additionally, we detected strong associations between bioactive green tea compounds in plasma and specific gut bacteria, including associations between spiramycin and *Gemmiger* and between wildforlide and *Anaerorhabdus*. Notably, some of the physiologically relevant green tea compounds are likely derived from plant-associated microbes, highlighting the importance of considering foods and food products as meta-organisms. Overall, we describe a novel workflow for discovering relationships between individual food compounds and the composition of the gut microbiome.

**IMPORTANCE** Foods contain thousands of unique and biologically important compounds beyond the macro- and micro-nutrients listed on nutrition facts labels. In mammals, many of these compounds are metabolized or co-metabolized by the community of microbes in the colon. These microbes may impact the thousands of biologically important compounds we consume; therefore, understanding microbial metabolism of food compounds will be important for understanding how foods impact health. We used metabolomics to track green tea compounds in plasma of mice with and without complex microbiomes. From this, we can start to recognize certain groups of green tea-derived compounds that are impacted by mammalian microbiomes. This research presents a novel technique for understanding microbial metabolism of food-derived compounds in the gut, which can be applied to other foods.

**KEYWORDS**  microbiome, metabolomics, 16S RNA, nutrition, multi-omics, symbiosis, gnotobiotic, food, polyphenols

Address correspondence to Nichole A. Reisdorph, nichole.reisdorph@cuanschutz.edu.

The authors declare no conflict of interest.

See the funding table on p. 17.

Assessing the impact of food on health is challenging (1). This is exacerbated by the unique co-metabolism of foods by the host and the gut microbiome across individuals with distinct microbiota. Owing to recent advances in omics technologies, determining the identities of microbial and microbial:host metabolites following the consumption of specific foods is now possible (1–3). For example, Wang et al. found that a product of microbiome:host co-metabolism of phosphatidylcholine, trimethylamine N-oxide (TMAO), predicted risk for cardiovascular disease (CVD) (4). In addition, integrated metabolomics and microbiome approaches are being applied with promising results (5–12). However, challenges remain, including identifying metabolites that are specifically produced by the microbiome or by host:microbiome interactions.

Alteration of the gut microbiome has been associated with the development of multiple disorders (13), including depression and anxiety (14), metabolic syndrome (15), and inflammation (16). A broad understanding of how the microbiome affects the host metabolome has come from comparing germ-free mice with those colonized with the microbiota of various humans (17) and/or treated with antibiotics compared with non-antibiotic treated controls (17–19). These studies found that microbial composition has a profound influence on the presence and relative abundances of many metabolites in various sites, including the blood, urine, feces, and the gastrointestinal tract. Additionally, dietary differences clearly influence microbial community structure, as certain substrates favor specific taxa. For example, Wu et al. (20) reported a greater relative abundance of *Bacteroidetes* with high animal protein consumption and a greater relative abundance of *Prevotella* with plant consumption, which is supported by other research investigating the effects of a Mediterranean diet intervention (21). Together, these findings suggest that unique microbiome profiles may be bidirectionally linked to specific dietary components.

Even dietary components that are not calorically dense, such as green tea (GT), have effects on mammalian health. Several studies have suggested that various components of tea, including flavonoids and phenolic acids, have both anti-inflammatory and neuroprotective effects (22–25). These polyphenols are metabolized by a combination of the host and microbiome, which means that variation in the functional capacity of the microbiome along the gastrointestinal tract will affect the downstream host metabolism of GT-derived compounds (26). However, the exact mechanisms by which the microbiome alters GT metabolism and downstream molecular networks are not yet understood. To our knowledge, only a few studies have focused on the metabolism of tea in the context of the microbiome. For example, Axling et al found that supplementing mice with GT powder along with *Lactobacillus plantarum* promoted the growth of *Lactobacillus* and attenuated high fat diet-induced inflammation (27). Studies utilizing humanized mice are a logical next step for determining the metabolites responsible for these effects and which microbes might be involved in their metabolism.

Our approach begins to address this challenge through the use of human colonized (HU) mice fed GT. It aims to determine how the gut microbiome affects which GT compounds are found in the plasma of HU and low complexity microbiome (LCM) mice. These scientific premises resulted in the formation of our **hypotheses** that specific GT compounds in plasma will associate with specific bacterial genera or be altered in concentration by the presence of a gut microbiome. We tested these hypotheses by colonizing gnotobiotic mice with microbiomes from 10 healthy humans and then treating both colonized and LCM mice with an acute dose of GT extract by oral gavage (Fig. 1).

Because diet is among the most significant modifiers of human health, a detailed understanding of the microbiome in the context of food metabolism is critical for disease modification and prevention.

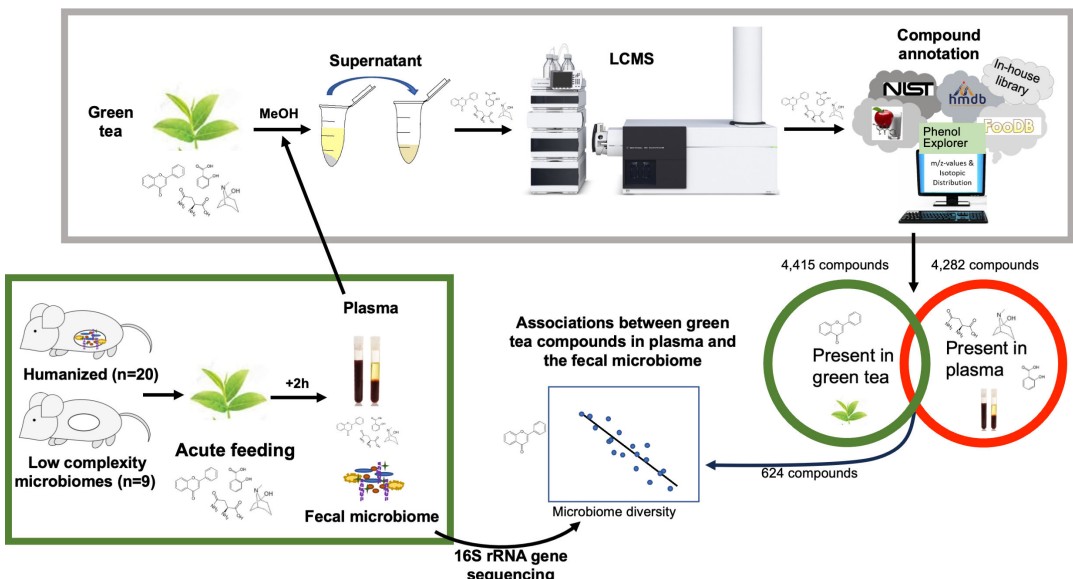

**FIG 1** Compounds from green tea were detected in plasma after green tea gavage in mice. Flowchart shows the study design. A total of 29 mice (20 humanized microbiomes and nine low complexity microbiomes) were gavaged with 100 µL of green tea then sacrificed 2 h later, and plasma, whole brain, and fecal microbiome samples were collected. Liquid chromatography-mass spectrometry-based metabolomics was performed on green tea samples and plasma. Plasma metabolomics data were mined to determine which green tea compounds could be found in plasma. Relationships between the fecal microbiome and green tea compounds found in plasma were assessed. Abbreviations: LCMS, liquid chromatography-mass spectrometry; MeOH, methanol; rRNA, ribosomal ribonucleic acid.

## MATERIALS AND METHODS

### Chemicals, standards, and reagents

Chemicals, standards, and reagents used for sample preparation and liquid chromatography-mass spectrometry (LCMS) analysis were of high-performance liquid chromatography (HPLC) or LCMS grade. These included water from Honeywell Burdick & Jackson (Muskegon, MI, USA), methanol, and J.T. Baker methyl tert-butyl ether (MTBE) from VWR (Radnor, PA, USA), formic acid from ThermoFisher Scientific (Waltham, MA, USA), Fisher Chemical acetonitrile and methanol from Fisher Scientific (Fair Lawn, NJ, USA), and Supelco 2-Propanol from Millipore Sigma (Burlington, MA, USA). Authentic standards for sample preparation were from Avanti Polar Lipids Inc. (Alabaster, AL, USA), Cambridge Isotope Laboratories (Tewksbury, MA, USA), Sigma-Aldrich (St. Louis, MO, USA), and CDN Isotopes (Pointe-Claire, Quebec, Canada).

### Human fecal sample collection

Human fecal samples were collected from 10 healthy participants using a commode specimen collector. Fecal samples were then shipped cold within 48 h, aliquoted, and stored at −80°C prior to use in gnotobiotic mouse experiments. Collection of the human fecal samples used in these experiments was approved by the Colorado Multiple Institutional Review Board (COMIRB) COMIRB #14–1595.

### Mouse colonization

All mouse studies were approved by the University of Colorado Institutional Animal Care and Use Committee (IACUC). C57BL/6 germ-free mice from the colony maintained at the University of Colorado Gnotobiotics Core were placed on an irradiated, low polyphenol diet (TD.97184, Envigo, Indianapolis, IN) at the time of weaning (3 weeks of age) through 6 weeks of age. At 4 weeks of age, mice were orally gavaged with 200 µL of either human fecal slurry (100 mg stool homogenized in 1 mL reduced phosphate-buffered saline [PBS]

in an anaerobic chamber) or PBS for LCM controls ($n$ = 9). A total of 20 mice received 10 distinct human fecal slurries, for an $n$ = 2 mice per human gut microbiome. Due to the limited availability of LCM mice, the experiment was conducted in three cohorts as described in the Supplementary Methods section. Mice were housed individually in a sterile caging system to maintain their LCM or individualized colonization status for the remainder of the experiment. Mice were weighed three times weekly and on the day of GT dosing.

## Mouse GT gavage

Gavage solutions of GT were prepared under sterile conditions in a Nuair Biological Safety Cabinet (Plymouth, MN). LCMS water was heated to 70°C and sterile filtered through a Steriflip disposable vacuum filtration tube (Merck KgaA, Darmstadt, Germany). Jade Leaf brand matcha green tea ("Culinary Grade Premium Second Harvest - Authentic Japanese Origin [8.8 Ounce Pouch]," Seattle, WA) was irradiated by Envigo at the same time as the low polyphenol diet and stored in a sterile 50 mL Falcon Tube (Corning, Glendale, AZ) at 4°C. GT slurries were prepared at 10 mg/mL with warm sterile water, vortexed for 15 s to mix, and then stored at 4°C until gavage. A total of 29 mice (20 HU, 9 LCM) were gavaged with 100 µL of GT slurry (50 mg/kg equivalent). Four leftover slurries were stored at −80°C for GT metabolomics analysis.

## Plasma and tissue collection

Mice were sacrificed approximately 2 h after the GT gavage. Blood was collected in 1.3 mL K3 EDTA micro sample tubes (Sarstedt Inc., Nuembrecht, Germany) via sub-mandibular bleeding, inverted five times and immediately placed on ice. Blood was centrifuged at 3,000 × $g$ for 30 min at 4°C, within 30 min of collection. Plasma was aliquoted into 1.5 mL microcentrifuge tubes (Fisher Scientific) and stored at −80°C until analysis. Fecal microbiome samples were collected in 1.5 mL microcentrifuge tubes, flash frozen in liquid nitrogen, and stored at −80°C until analysis. The 2 h time point was selected following a small pilot utilizing wild-type (WT) mice whereby GT compounds were detected in higher numbers and abundance compared with 4 and 24 h time points (Supplementary Methods section).

## Metabolomics

Frozen plasma samples and four GT extracts were thawed on ice and prepared for analyses as previously described (28–30). Briefly, proteins were precipitated and small molecules were extracted from supernatants using an organic liquid-liquid extraction technique with methyl tert-butyl ether (MTBE) and water. This resulted in hydrophilic (aqueous) and hydrophobic (lipid) fractions. Nine hydrophobic and six hydrophilic labeled spike-in standards were used to monitor instrument performance and sample preparation batch variability (31). Spiked and un-spiked methanol preparatory blanks were prepared alongside study samples. Quality control (QC) included the analysis of purchased plasma (BioIVT, Westbury, NY), spiked with authentic standards and prepared alongside study plasma samples (Supplementary Methods section).

Aqueous and lipid fractions were analyzed by LCMS using published methods (29, 31, 32). Briefly, the hydrophobic fraction was analyzed using reverse phase C18 chromatography and a quadrupole time-of-flight mass spectrometer (QTOF 6545, Agilent Technologies, Santa Clara, CA) in positive ionization mode. The hydrophilic fraction was analyzed using an SB-Aq column (Agilent Technologies) on the same instrument. LCMS method details are provided in the Supplementary Methods section.

## Metabolomics data processing

Metabolomics spectral data for both plasma and GT extracts were processed using a recursive workflow and area-to-height conversion with Agilent's MassHunter Profinder ver. 10.0 service pack 1 (Profinder) and Mass Profiler Professional Ver. 15.1 (MPP, Agilent

Technologies) (10). Data from lipid and aqueous fractions were extracted separately in Profinder using Batch Molecular Feature Extraction followed by Batch Targeted Feature Extraction. Compounds found in blanks were removed. Plasma samples were limited to compounds eluting before 10.4 and 13 min for lipid and aqueous fractions, respectively. GT samples were limited to compounds eluting before 10.4 and 11 min for lipid and aqueous fractions, respectively. Compounds eluting past these times had poor signal-to-noise ratios and were not used.

Plasma compounds were annotated using MassHunter ID Browser ver. 10.0 (Agilent Technologies) by searching custom in-house databases and public databases consisting of compounds from METLIN, Lipid Maps, Kyoto Encyclopedia of Genes and Genomes, and Human Metabolome Database using accurate mass and isotope ratios. An initial database search was conducted using H+ as the primary charge carrier. Unannotated compounds were re-searched using the same databases with loss of water included. These compounds were designated Metabolomics Standards Initiative level three and are listed by software-assigned compound number (e.g., C1287 is compound number 1287) (11). Plasma compound lists were exported for statistical analysis and to determine which GT compounds were detected in plasma.

## Green tea compounds in plasma

Data from the four GT extracts were processed as above with the following adjustments: GT compounds had to be present in 100% of GT samples, and the compounds were restricted to those eluting before 10 min for both the aqueous and lipid fractions. In addition to the databases used to annotate plasma, the GT samples were also searched against Phenol Explorer, FooDB, and a Traditional Chinese Medicine natural products database (Agilent Technologies). The GT compound lists were exported as .cef files and imported into Quantitative Analysis (Agilent Technologies).

To verify that GT compounds originated from GT, data from a separate experiment using mice not fed green tea were used to determine baseline differences (Supplementary Methods section). Briefly, 10 mice were gavaged with the same 10 distinct human fecal slurries as described above (i.e., one microbiome per mouse), and six mice were gavaged with PBS. At 2 weeks post-gavage, plasma was collected as described in the main text. LCMS data for plasma samples were processed as described above and imported into Quantitative Analysis. Extracted ion chromatograms were generated for each GT compound from plasma LCMS data using a "targeted data extraction" strategy. The method parameters included the following: left and right extraction window of 10 ppm, retention time extraction window ±0.2 min, and retention time outlier >0.3 min. For each GT compound, the monoisotopic peak (M1) and C13 peak (M2) were used for qualification. The M1 peak was used as the quantifier ion, whereas M1 and M2 peaks (or only M1 when no M2 peak was detected) were used as qualifier ions. When a second charge carrier such as sodium was detected, its monoisotopic peak was used as a qualifier. Qualifier relative uncertainty was set to 20%. The analysis results were imported into MPP and filtered to retain only targets that were present in at least three plasma samples with an area ≥20,000 counts for the aqueous fraction and ≥55,000 counts for the lipid fraction. These compound lists, including area under the curve (AOC) for relative quantitative comparison, were exported for statistical and informatic analysis.

Whole metabolomics data sets were visualized using principal components analysis for the purpose of quality control (Supplementary Methods section).

## Microbiome–DNA extraction and sequencing

DNA was extracted from the fecal samples using the Power Soil Pro Kit protocol (Qiagen, Germantown, MD). Barcoded primers targeting the V4 region of the16S rRNA gene were used to amplify the extracted bacterial DNA via polymerase chain reaction (PCR) using the Earth Microbiome Project (EMP) standard protocols (http://www.earthmicrobiome.org) (33). PCR product quantification was completed using PicoGreen (Invitrogen, Carlsbad, CA) and used to pool equal amounts of amplified DNA from each sample. The

QIAquick PCR Purification Kit (Qiagen) was used to clean the pooled libraries. Three runs were used to generate sequences using the Illumina MiSeq platform (San Diego, CA).

## Microbiome–sequence data processing

Microbiome sequencing data were processed in QIIME2-2021.8 (34). Sequences were demultiplexed using the q2-demux plugin and then denoised using q2-dada2 (35) with a truncation length of 230 bp. A phylogenetic tree was created using Saté-enabled phylogenetic placement via the q2-fragment-insertion plugin (36). Taxonomy was assigned to reads using a naive Bayes classifier trained on the latest Greengenes 1 database as of January 2022 (Greengenes 13.8) (37). Reads with no taxonomy below the kingdom level and reads classified as mitochondria and chloroplast were removed from further consideration. For diversity analyses, the samples were rarefied to 40,625 reads per sample, which was the highest rarefaction depth that did not exclude any samples. A taxa bar plot was created using Microshades (38).

## Differential abundance testing

Analysis of compositions of microbiomes with bias correction (ANCOM-BC) (39) was used to assess differentially abundant taxa between LCM and HU mice. ANCOM-BC was performed at the family and genus level, and a Benjamini-Hochberg FDR correction was applied to $P$ values.

## Diversity metrics

Alpha diversity was calculated using Faith's phylogenetic diversity (40), and microbiome beta diversity was calculated using unweighted UniFrac distances (41). Metabolome beta diversity was calculated using the Bray-Curtis distance metric.

## Ordination

Microbiome principal coordinate analysis (PCoA) plots were generated using the unweighted UniFrac distance matrix via the Python package scikit-bio (v0.5.6) (42), and metabolomics PCoA plots were generated using Bray-Curtis distances.

## Calculation of centroids for distance-based statistics

To avoid pseudoreplication (43) affecting precision of estimates in permutation-based tests where random effects or cluster-robust standard errors cannot be used, we collapsed microbiome UniFrac distances to centroids within humanized microbiome donors or LCM centroids within each experimental cohort using the dist_multi_centroids function from the usedist R package (v0.4.0) (44). This reduced sample size from $n = 29$ (20 mice with microbiomes humanized in pairs from 10 unique donors + 9 LCM across three experimental cohorts) to $n = 13$ (10 averaged HU centroids corresponding to each donor microbiome + 3 LCM centroids corresponding to each cohort). Corresponding non-distance data (such as compound abundance) were averaged (arithmetic mean) within groups.

## Procrustes randomization test

Procrustes randomization tests were performed on the metabolomics and microbiome PCoA coordinates from centroid distances using the protocol described by Peres-Neto and Jackson (45). Specifically, we first performed a Procrustes transformation on the observed data sets and then permuted the GT lipid or GT aqueous compound Bray-Curtis distance PCoA coordinates $10^4$ times and performed a Procrustes transformation on the permuted data sets. The $P$ value was calculated based on the portion of permuted Procrustes-transformed data sets with resulting $m^2$ (Gower's statistic, also referred to as disparity) scores lower than the Procrustes $m^2$ score of the observed data sets.

## Compounds associated with microbiome composition

To identify compounds and neurochemicals associated with microbiome community composition, PERMANOVA tests were performed using Adonis2 from the R package vegan (v2.6–4) (46) to assess correlations between plasma abundances of each individual compound and the microbiome unweighted UniFrac distance matrix (using $10^4$ permutations). *P* values were adjusted with a Benjamini-Hochberg correction to reduce false discovery rates (FDR). We used the R package Selbal (v0.1.0) (47) with 10-fold cross-validation to identify genus-level balances that explained compound abundance.

## Compounds associated with microbiome diversity

We performed linear regression with cluster robust standard errors on each individual compound and Faith's phylogenetic diversity to assess which compounds were associated with microbiome diversity. Standard errors were clustered by humanized microbiome donor or by experiment for LCM mice, and *P* values were calculated based on clustered standard errors using the R package fixest (v0.11.2) (48). *P* values were adjusted with a Benjamini-Hochberg correction to reduce FDR. Regressions with any data points that had a dffit (difference in model fit if the point was removed) absolute value greater than one were removed from further consideration.

## Mixed-effects multi-omics modeling

Associations between metabolites and microbes were assessed by pairwise linear mixed effects models between each microbe and each metabolite, with the following formula:

$$z\ score(metabolite)\ \sim\ \beta * arcsinh(microbe\ *\ 100)\ +\ (1\,|\,humanized\ id)\ +\ \epsilon \quad (1)$$

Metabolite concentrations were z-score transformed so that $\beta$ coefficients could be compared across compounds. Microbe relative abundances were multiplied by 100 to transform to percent relative abundance and then arcsinh transformed $\left(arcsinh(x) = ln\sqrt{x^2 + 1}\right)$ to alter the distributions of relative abundances to minimize violations of linear modeling assumptions across features. (1|*humanized id*) indicates a random effect for the humanized microbiome ID or LCM status. The maximum influence of any data point was calculated for each regression using the mdffits function from the R package HLMdiag (v0.5.0) (49). Any regressions with any data points that had an influence (difference in model fit if the point was removed) absolute value greater than or equal to four were removed from further consideration. Models were run using the R package lme4 (v1.1–29) (50), and *P* values were calculated based on Satterthwaite's degrees of freedom using the R package LmerTest (v3.1–3) (51). *P* values were adjusted with a Benjamini-Hochberg correction to reduce FDR.

## RESULTS

### Metabolomics of mouse plasma and GT

Metabolomics of the mouse plasma resulted in 4,282 lipid and aqueous compounds, whereas metabolomics of GT extract resulted in 4,415 compounds (Fig. 1). Of those, 624 GT compounds were detected in at least three plasma samples following GT gavage. In total, 432 were found in the lipid-rich extract (GT lipids), whereas 192 were found in the aqueous extract (GT aqueous).

We removed from consideration any plasma GT compounds that were also found in the plasma of mice not gavaged with GT. This step removed 86 lipid compounds and 59 aqueous compounds, resulting in a total of 145 compounds removed. After all filtering steps, 479 (346 lipids, 133 aqueous) compounds remained, which were evaluated for their relationships with microbiome composition and diversity. These compounds are listed in the Supplementary Data.

## Contamination of germ-free controls

Following 16S rRNA gene amplicon sequencing of LCM and HU fecal microbiomes, it was discovered that the LCM mice were not germ-free as intended and instead had microbiomes (Fig. 2). Additionally, the LCM controls had small cecums, which are uncharacteristic of germ-free mice (data not shown). Specifically, the LCM controls were dominated by 6 Amplicon Sequence Variants (ASVs) in the Firmicutes phylum, which made up over 90% of the detected reads. Table S1 shows the taxonomic assignment and relative abundances of these ASVs. Following extensive investigation, it was determined that the contamination resulted from the irradiated low polyphenol food that was fed to mice. For example, bacterial colonies grew when irradiated food was plated, and PCR analysis showed a range of 3.5 –4.6 ng/µL in the three irradiated samples tested. No growth was seen when irradiated GT was plated. Discussions with the vendor were largely inconclusive, but it was surmised that the source of bacteria may have been casein used in the diet (52). Although not ideal, because all mice in the study received the same food, it was determined that all mice were exposed to this contamination, and therefore, we continued with data analysis, focusing on statistical methods that consider microbiome community composition or diversity rather than solely the group status (LCM vs HU).

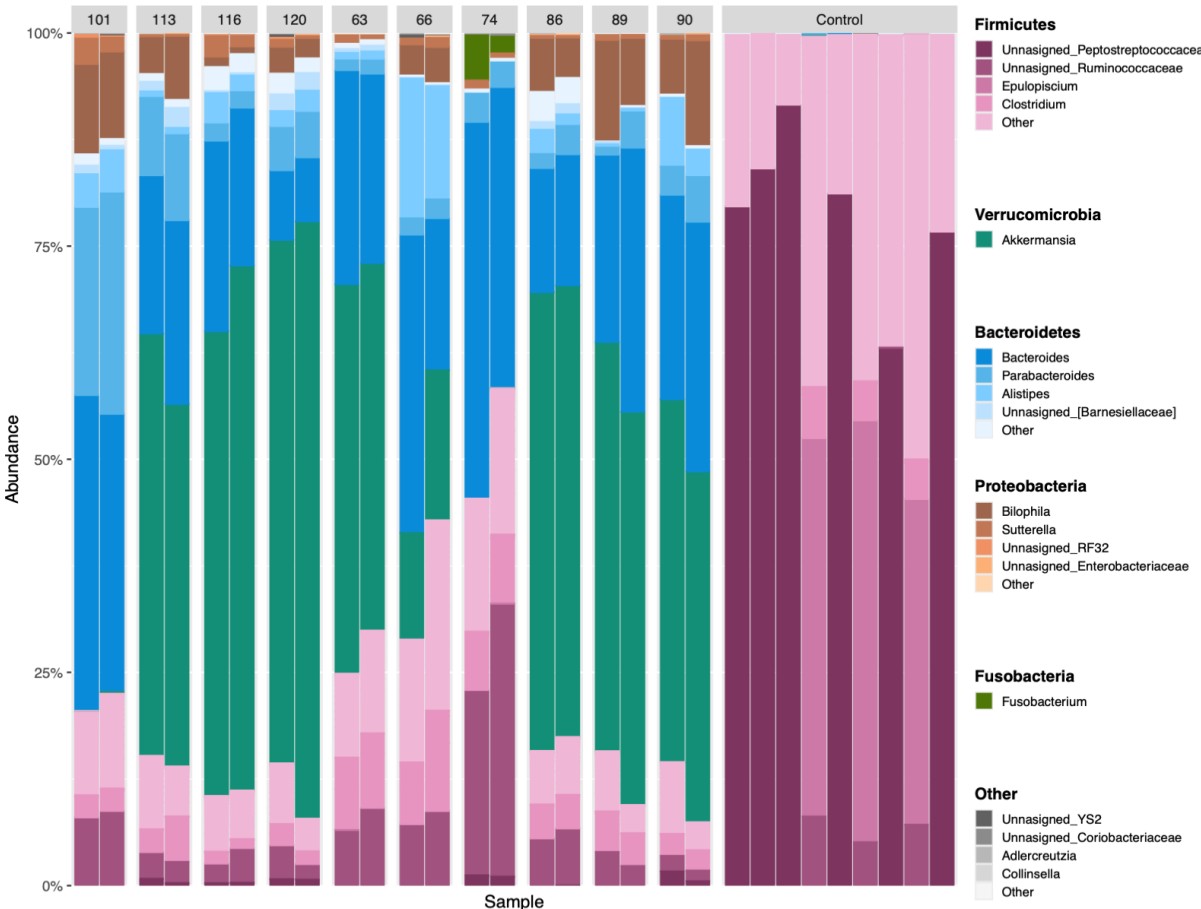

**FIG 2** Taxa bar plot shows the differences in phylum and genus level composition between low complexity microbiome and humanized microbiome mice. Bars are colored by phylum and microshaded by genus, and the height of each colored bar represents the relative abundance of the corresponding taxon. One stacked bar represents the composition of one mouse's microbiome, and humanized microbiome IDs are shown along the x-axis. Low complexity microbiome mice are labeled as "control" and have vastly different compositions compared with humanized mice.

## Taxonomic composition of fecal microbiomes

LCM and HU mouse microbiomes differed in community composition (pseudo-$F$ = 5.9, $P$ = 0.005). LCM mice were colonized almost entirely by Firmicutes, with six of the nine mice having high levels of unassigned ASVs in the family Peptostreptococcaceae (Fig. 2, $P$ = 0.005). Additionally, LCM mice had higher relative abundances of *Epulopiscium* (ANCOM-BC; $P$ = 0.015) and *Turicibacter* (ANCOM-BC; $P$ = 0.002) in their microbiomes compared with HU mice (Fig. 2).

Microbiome composition was very similar within HU mouse replicates (UniFrac distance within pairs mean [95% CI] = 0.25 [0.21–0.30]), and variable between mice colonized with different human donors (UniFrac distances between centroids mean [95% CI] =0.50 [0.49–0.52]) (Fig. 2). This is supported by a PERMANOVA, demonstrating larger variance in phylogenetic composition between pairs than within pairs ($R^2$ = 0.87, psuedo-$F$ = 7.6, $P$ < 0.001). The microbiome of all but two pairs of HU mice had *Akkermansia* present at high relative abundances (≥20%). All HU pairs had representation of Bacteroidetes, primarily including *Bacteroides*, *Parabacteroides,* and *Alistipes*. Proteobacteria were present in the microbiome of all HU but not LCM mice, and this phylum predominantly consisted of ASVs belonging to *Bilophila* and *Sutterella*. Some HU pairs also had low relative abundances (<5%) of *Fusobacteria* in their fecal microbiome.

## Diversity, composition, and relationships between the fecal microbiome, plasma metabolome, and plasma lipidome

Both alpha and beta diversities of the fecal microbiome differed between HU and LCM mice. Specifically, microbiome phylogenetic composition (UniFrac distance) strongly differed between LCM and HU mice (pseudo-$F$ = 5.9, $P$ = 0.005), and clear separation between LCM and HU can be seen in the unweighted UniFrac PCoA (Fig. 3A). Additionally, Faith's phylogenetic diversity was significantly higher in HU mice (mean [95% CI] = 12.0 [12.3–13.3]) than LCM mice (mean [95% CI] = 4.1 [2.3–5.9]; regression $\beta$ = 8.7, $P$ < 0.001; Fig. 3B).

The overall composition of GT-specific lipids in the plasma was not different between LCM and HU mice (Bray-Curtis distance PERMANOVA on centroids $R^2$ = 0.12, pseudo-$F$ = 1.5, $P$ = 0.14; Fig. 3C), nor was the composition of aqueous GT-specific (Bray-Curtis distance PERMANOVA on centroids $R^2$ = 0.16, pseudo-$F$ = 0.17, $P$ = 0.96; Fig. 3D). This can be seen in PCoA plots, where the GT compounds in plasma do not cluster according to LCM or HU group (Fig. 3C and D). The portion of shared GT compounds present in plasma was high in both the lipid and aqueous data sets (Jaccard similarity lipid mean [min – max] = 1.0 [0.99–1.0]; aqueous mean [min – max] = 0.95 [0.88–0.98]. Furthermore, a Procrustes randomization test did not reveal an overall significant relationship between microbiome composition (UniFrac) and GT lipids in plasma (Fig. 4A, $m^2$ = 0.39, $P$ = 0.085) or aqueous GT compounds in plasma (using Procrustes-transformed Bray-Curtis PCoA, Fig. 4B, $m^2$ = 0.66, $P$ = 0.79).

## Green tea compounds associated with microbiome composition

One goal of the study was to determine if GT compounds in plasma associate with microbiome diversity and composition. Understanding these associations is a first step toward determining which bacterial species may be responsible for the metabolism of specific GT compounds. PERMANOVA tests on centroids assessing associations between the concentration of each GT compound and microbiome composition (unweighted UniFrac) found no significant relationships after FDR correction. For reference, Table S2 shows compounds with uncorrected $P$ < 0.05.

The lack of significance looking while comparing multivariate similarity between samples may be too unfocused. It is very likely that many compounds would be unaffected by most members of the microbiome, as many food-derived compounds are absorbed in the small intestine. We thus looked for finer-grained relationships between individual compounds and smaller sets of microbes.

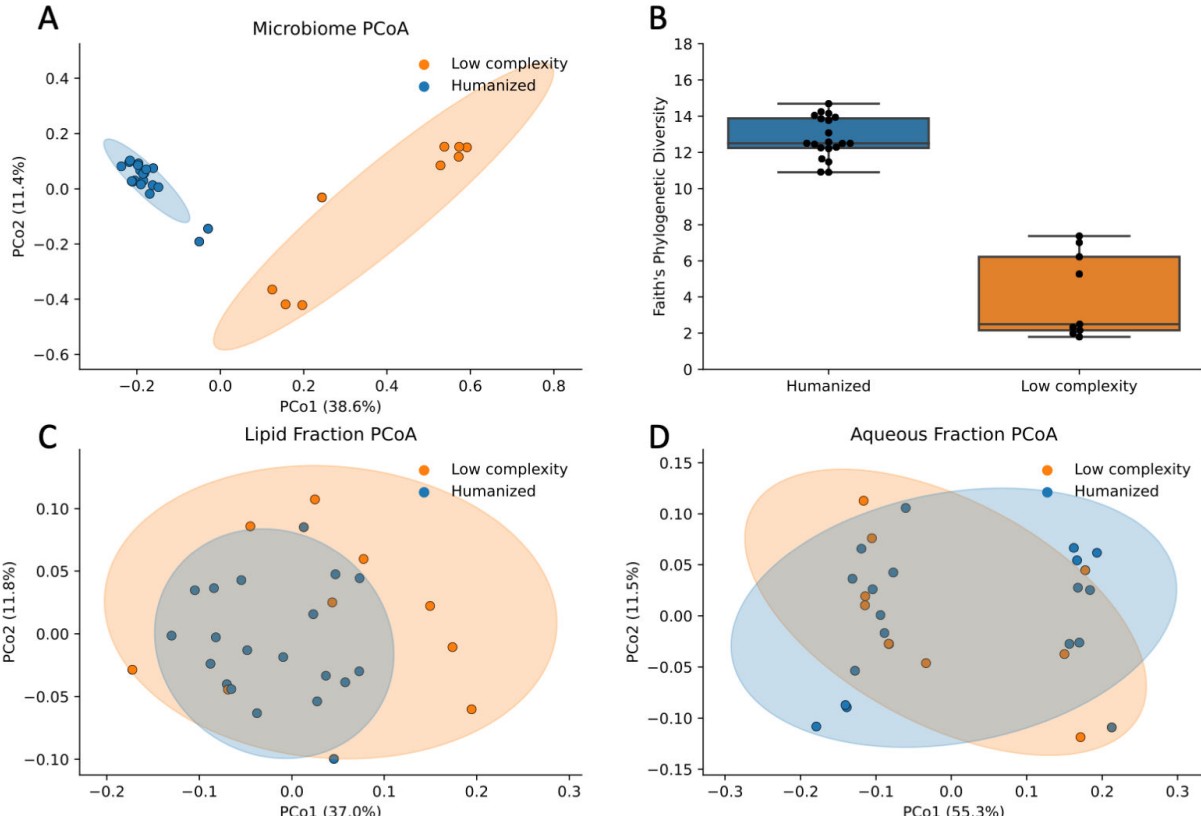

**FIG 3** Microbiome PCoA and alpha diversity show a clear separation between low complexity microbiome and humanized mice, whereas metabolome PCoA plots do not separate as clearly. (A) Microbiome unweighted UniFrac PCoA is colored by humanized vs low complexity microbiome status. (B) Boxplot shows higher Faith's phylogenetic diversity in humanized vs low complexity microbiome mice. (C) PCoA of the lipid fraction of green tea compounds in plasma is colored by humanized vs low complexity microbiome status. (D) PCoA of aqueous fraction of green tea compounds in plasma is colored by humanized vs low complexity microbiome. Abbreviation: PCoA, principal coordinates analysis; PCo, principal coordinates axis.

Although whole community-level associations were not found, for the 12 compounds most associated with microbiome composition, Selbal-identified balances were able to explain 45%–70% of the variation in the abundance of these compounds, and all had a regression slope-adjusted $P < 0.05$. Gamma-glutamyl-alanine having the most variation explained by a positive correlation with a balance of the genera *Alistipes* to *Butyricimonas* (Fig. 5). Balances identified by Selbal are ratios of taxa, meaning that as gamma-glutamyl-alanine increased, *Alistipes* increased relative to *Butyricimonas,* or *Butyricimonas* decreased relative to *Alistipes*.

### Green tea compounds associated with microbiome phylogenetic diversity

When testing GT compound associations with microbiome alpha diversity (Faith's phylogenetic diversity), eight compounds had FDR-corrected $P < 0.05$ (Fig. 6; Table S3), including lactiflorin, acetylagmatine, aspartic acid, 7,8- dihydroparasiloxanthin, montecristin, and CL (82:16) (Fig. 7). Most of these compounds had negative associations with microbiome diversity, except for CL (82:16) and an unannotated compound.

### Relationships between individual microbes and GT compounds

When controlling for the humanized microbiome with which mice were colonized with a random effect and applying a Benjamini-Hochberg FDR correction, across all mice (including LCM mice), we identified 161 significant relationships between microbes and GT compounds in plasma of the potential 33,433 relationships tested. Of these, the strongest associations were between an ASV in the genus *Bifidobacterium* and an

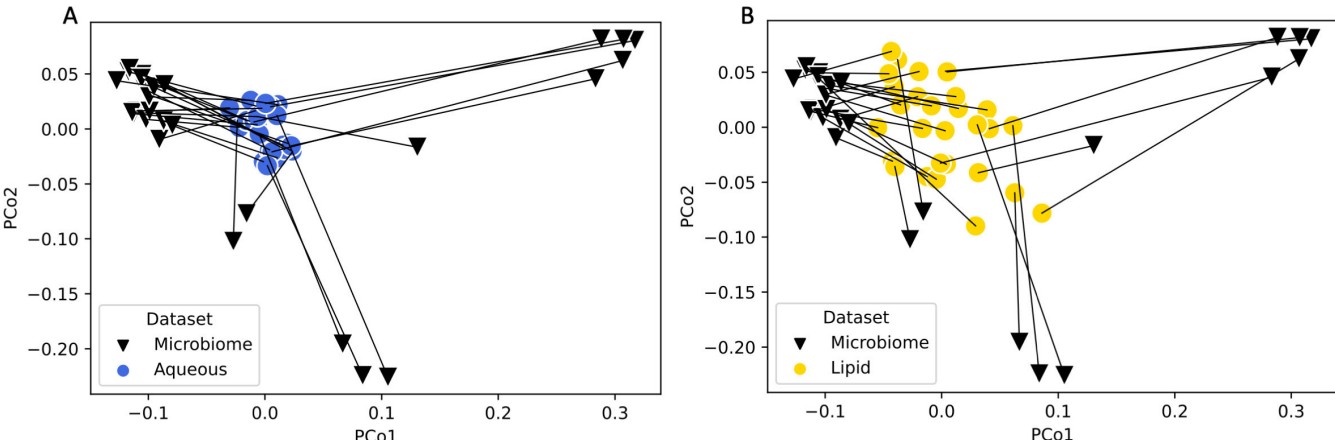

**FIG 4** Procrustes analysis does not indicate multivariate-multivariate relationships between green tea lipids or compounds in plasma and phylogenetic composition of the microbiome. Panel A shows the connection between microbiome composition and the composition of aqueous GT compounds detected in plasma, where each triangle represents the composition of one microbiome sample, and a line connects to that sample's corresponding aqueous GT compounds in plasma. Panel B shows the connection between microbiome composition and the composition of lipid GT compounds detected in plasma, where each triangle represents the composition of one microbiome sample, and a line connects to that sample's corresponding lipid GT compounds in plasma. Procrustes randomization test did not reveal an overall significant relationship between microbiome composition (UniFrac) and GT lipids in plasma ($m^2 = 0.39$, $P = 0.085$) or aqueous GT compounds in plasma ($m^2 = 0.66$, $P = 0.79$). In both panels, unweighted UniFrac microbiome PCoA coordinates were left untransformed, and metabolomics sample Bray-Curtis PCoA coordinates were transformed with a Procrustes function. Samples of the same type positioned close together were compositionally similar, whereas samples far apart were dissimilar. Shorter lines between samples indicate better overlap between the microbiome and metabolomics data sets. For statistical testing, samples belonging to mice from the same donor microbiome or low complexity microbiome mice from the same experimental cohort were collapsed into one centroid per treatment unit. Abbreviation: PCoA, principal coordinates analysis; PCo, principal coordinates axis.

unannotated compound, an ASV in the genus *Allobacterium* and montecristin ($\beta = 277.8$, $P < 0.001$, data not shown), and an ASV in the family Peptostreptococcaceae and acetylagmatine ($\beta = 0.5$, $P < 0.001$, data not shown).

Despite including LCM status as a covariate in each compound-taxon regression, there was still a large signal related to the high relative abundance taxa in the LCM mice, such as Peptostreptococcaceae. Thus, we also performed this analysis in only HU mice, as differences in the overall architecture of the LCM vs HU mice fecal microbiomes may have biased these individual metabolite-taxon relationships and violated the random effects assumption. When mixed effects regressions were performed for only HU mice, several GT compounds were significantly associated with specific bacterial ASVs. Of 30,594 potential relationships tested, 22 were significant after Benjamini-Hochberg FDR correction (Fig. 8A). For example, the GT compound wilforlide was positively associated with an ASV in the genus *Anaerorhabdus* ($\beta = 15.1$, $P < 0.001$). Spiramycin was positively associated with taxa in the genera *Gemmiger* ($\beta = 13.7$, $P < 0.001$) and *Lactobacillus* ($\beta = 12.9$, $P < 0.001$, Fig. 8B). Although multiple unannotated compounds had multiple significantly associated microbes, spiramycin was the annotated compound with more than one significant relationship with individual taxa (after FDR correction). The genus *Gemmiger* had the highest number of significant relationships with GT compounds, although three of these four relationships were primarily with unannotated compounds, and the relationships between compounds are not known.

## DISCUSSION

We show that GT compounds significantly associated with specific gut bacterial genera following the acute feeding of GT extract to mice. Our analysis strategy included an important data reduction step, whereby analysis focused on only GT compounds that were found in the plasma of mice who had consumed GT. This greatly increased power (via decreasing the need for as stringent FDR correction) by limiting analysis from 4,282 plasma compounds to 479 compounds that were specific to the intervention.

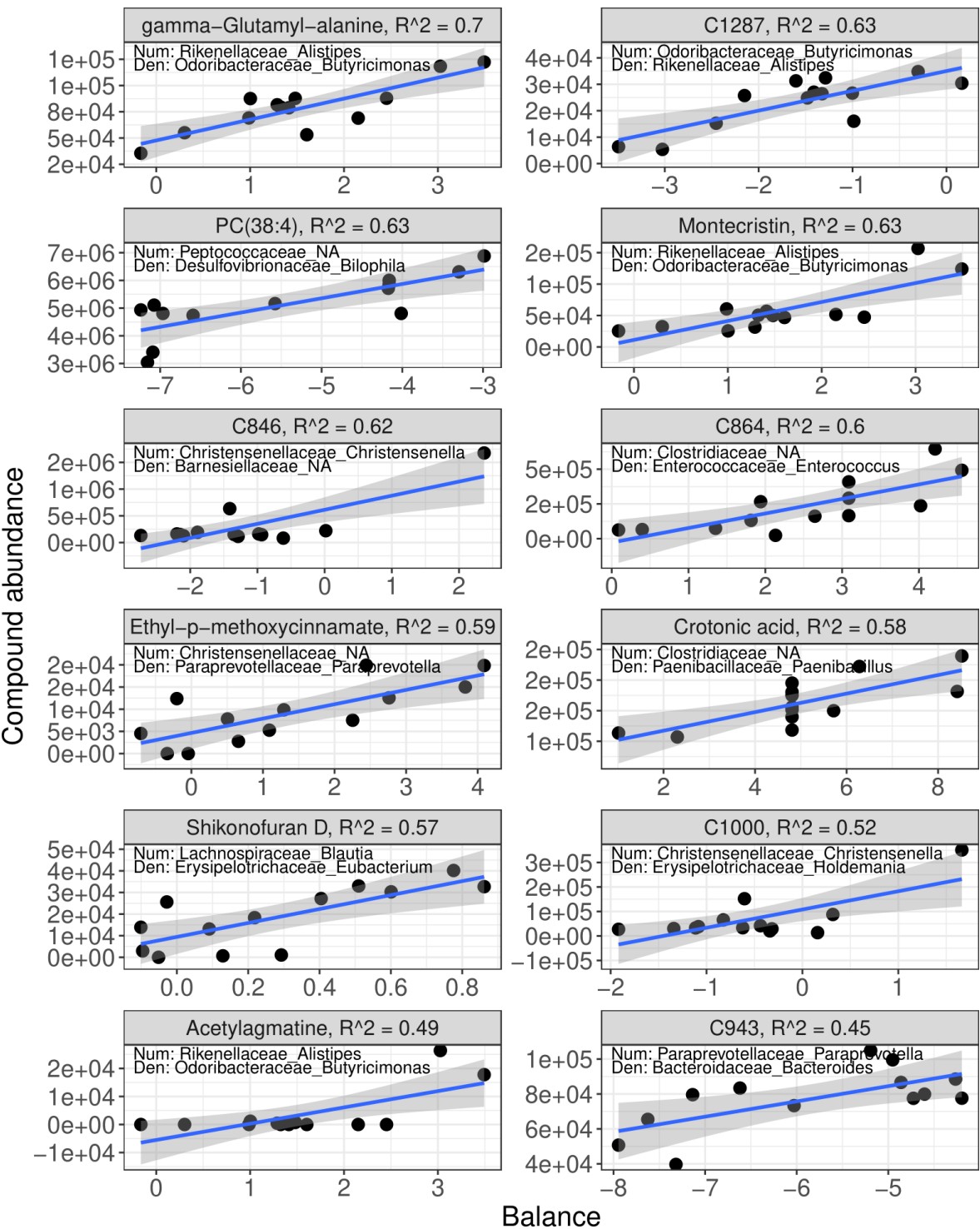

**FIG 5** Compositional balances in microbial relative abundances explain green tea compound abundance in plasma. Selbal with cross-validation was used to identify balances (log-ratios) of taxa that predict compounds that were associated with microbiome composition. The x-axis of each plot shows the microbiome balance, with the numerator and denominator labeled on the plot as <family>_<genus>. The y-axis shows the abundance of each green tea compound, which is labeled at the top of each plot, along with the regression $R^2$. The line on each plot shows the regression line of best fit and the shaded region indicates the 95% CI for that regression line. Each point represents the average balance for each donor microbiome or experimental control.

Importantly, these are compounds detected in dietary formulations of green tea, not solely compounds produced by the plant from which green tea is derived, *Camellia*

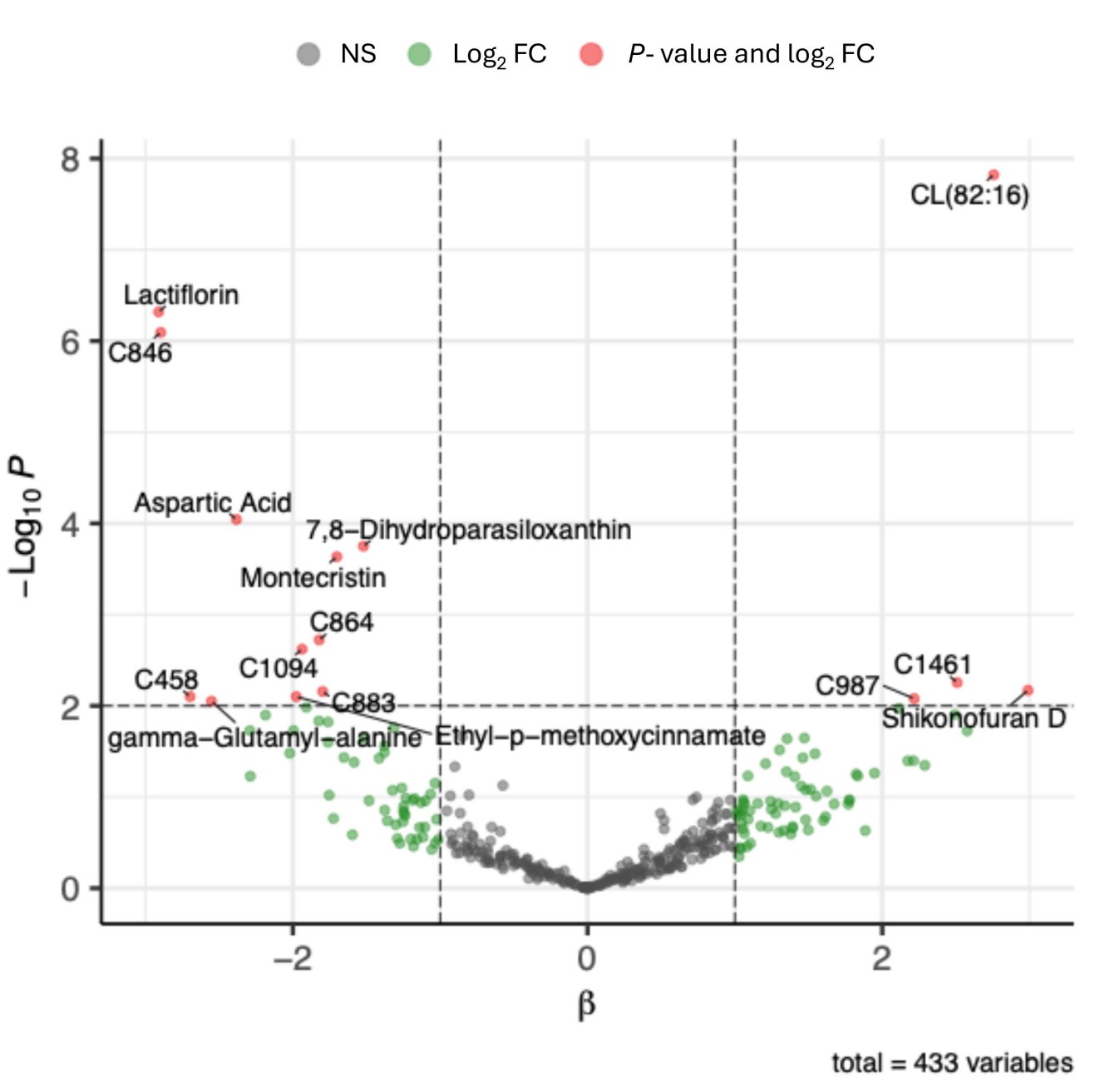

**FIG 6** Several green tea metabolites are associated with phylogenetic diversity of the gut microbiome. Each point represents one green tea compound. The y-axis represents *P*-values from a regression between that compound and microbiome phylogenetic diversity, and the x-axis represents the effect size of that compound's association with microbiome diversity. Compounds abundances were z-score transformed to allow for comparison of effect sizes, and *P*-values were calculated using cluster robust standard errors, where cluster ID was the donor microbiome source or experimental cohort for low complexity microbiome mice. The horizontal line represents $P = 0.01$, and vertical lines represent an effect size of 1, indicating that a 1 standard deviation (SD) increase in compound abundance was associated with a 1 unit increase in Faith's phylogenetic diversity. Compounds with $P < 0.01$ and fold change absolute value >2 are labeled with their annotation.

*sinensis*. Much like the mice in this study with associated microbiomes, *C. sinensis* plants are meta-organisms that coexist with microbial communities (53, 54). Microbial metabolites and even excreted metabolites from other nearby plants or pollinators may be

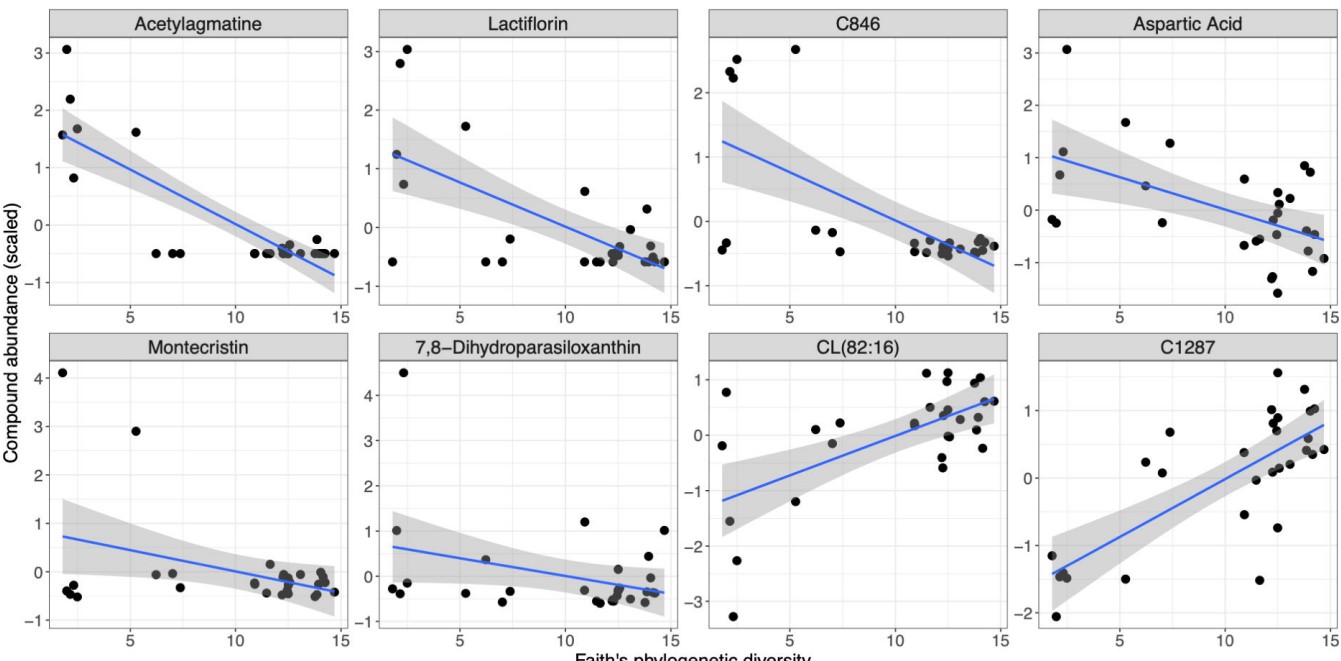

**FIG 7** Phylogenetic diversity of the gut microbiome is associated with green tea compound abundance in plasma. Fixed effects regression with cluster robust standard errors (clustered by humanized microbiome donor ID or low complexity microbiome experimental cohort) was used to identify green tea compounds that were associated with Faith's phylogenetic diversity of the gut microbiome, and compounds that had significant (Benjamini-Hochberg $P < 0.05$) relationship with Faith's phylogenetic diversity are shown. The x-axis of each plot shows Faith's phylogenetic diversity, and the y-axis shows the abundance of each green tea compound, which is labeled at the top of each plot. The line on each plot shows the regression line of best fit and the shaded region indicates the 95% CI for that regression line. Each point represents one sample.

present in the leaves used to create the tea used in this study; hence, the GT compounds discussed may come from a variety of sources across multiple kingdoms of life.

Sequencing of LCM and HU microbiomes from fecal pellets allowed us to perform analyses considering microbiome composition and diversity, rather than solely LCM status. This was particularly useful because the LCM mice in this study were populated with a low diversity microbiome due to contamination of the low polyphenol food used. Despite using phylogenetic composition of the microbiome in our analyses, we did not see multivariate-multivariate relationships between the overall composition of green tea compounds in plasma and the composition of the gut microbiome (per a Procrustes randomization test on PCoA coordinates from green tea compound Bray-Curtis distance and Unweighted UniFrac microbiome distance), which could be due to the absorption of many green tea compounds in the small intestine, before reaching the majority of microbial biomass in the colon.

Multiple compounds found in GT extracts were individually associated with microbiome composition and diversity. Importantly, these compounds are known to be present in plants and are relevant to mammalian physiology. For example, an annotated GT compound in plasma associated with microbiome diversity, montecristin, is an annonaceous acetogenin (a group of compounds that are waxy derivatives of fatty acids and are reported to have an array of health effects, including antimalarial, antiparasitic, and anti-cancer activities) (55). Another plant compound from GT that differed as a function of the microbiome was lactiflorin, a compound originally characterized in the plant genus *Paeonia*, that activates the antioxidant-controlling transcription factor nuclear factor erythroid 2-related factor (Nrf2) in rats (56). Lactiflorin, a monoterpene glycoside, has been used in the treatment of rheumatoid arthritis via inhibiting leukocyte recruitment and angiogenesis via, potentially, the VEGFR and PI3K-Akt signaling pathways, interleukin signaling, and platelet activation (57).

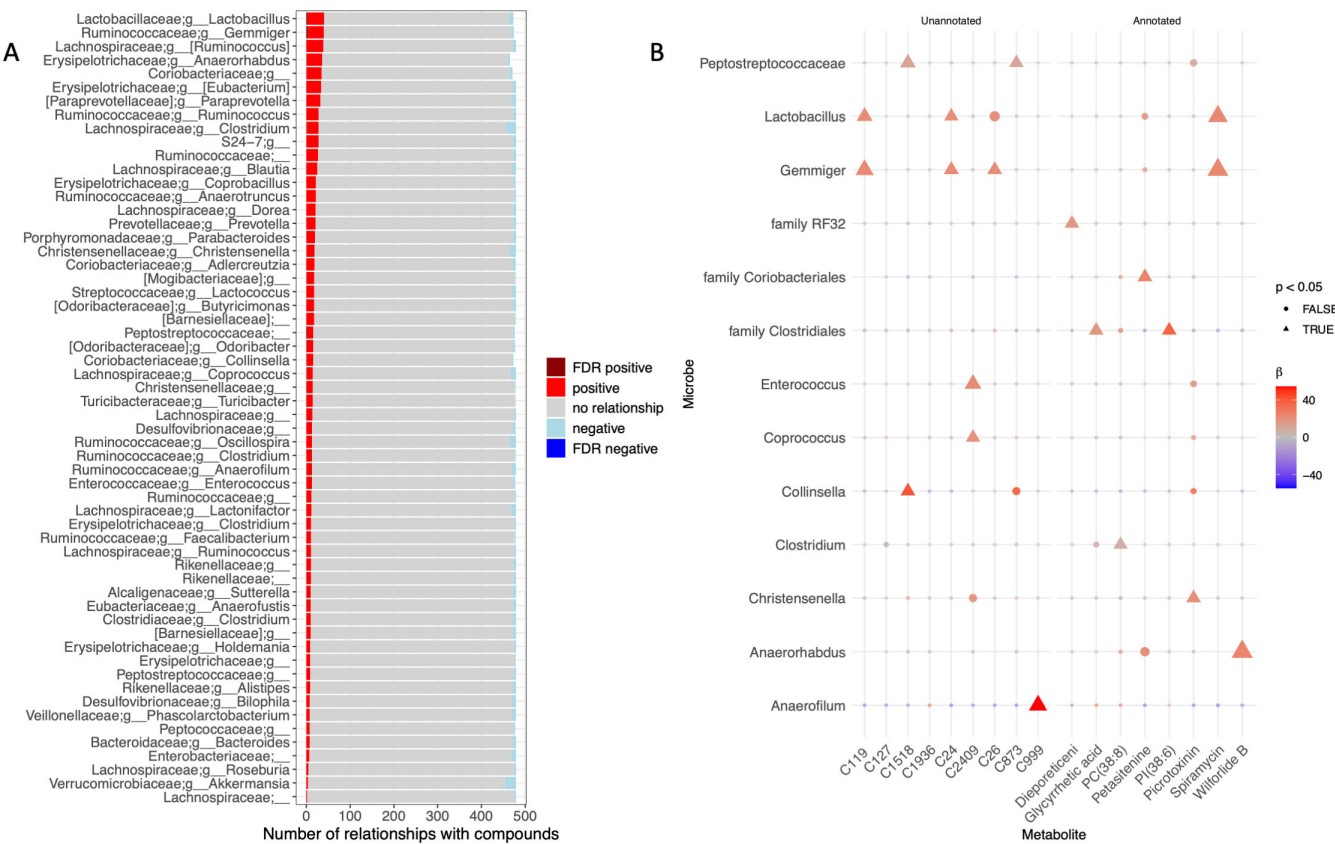

**FIG 8** Individual green tea compounds are associated with specific genera when controlling for overall microbiome composition. (A) A stacked bar plot shows a summary of relationships between green tea compounds and individual genus-level relative abundances, indicating the significance and direction of compound-microbe relationships. (B) A shaped heatmap shows the direction and significance of any microbes and compounds that had significant relationships after Benjamini-Hochberg p-value correction. To generate these results, 39,860 pairwise linear mixed effects regressions were run on z-score(metabolite) ~arcsinh(microbe) + (1|humanized_id), where (1|humanized_id) indicates the microbiome mice were humanized with, or if mice were in the low complexity microbiome group. Metabolites were transformed to have a mean of 0 and SD of 1 to allow for comparison of effect sizes across metabolites. Each point represents the regression run between the metabolite on the x-axis and the microbe shown on the y-axis. The color of each point represents the beta extracted from the model (coefficient of the microbe), and the size of the points represents the *P*-value of that coefficient. Circles represent *P* > 0.05, whereas triangles represent *P* < 0.05.

Previous studies have focused on the health effects of green tea catechins, a subgroup of flavonoids with known antioxidant properties (58). Although several catechins, including epigallocatechin 3-p-coumarate, catechin 3-gallate, gallocatechin 3'-gallate, and epigallocatechin 3-cinnamate were found in GT extract, these were not found at detectable levels in at least 3 mouse plasma samples and hence were not included in statistical analyses. This could be due to the concentration of extract used, the sensitivity of the assay, or the timing of blood draw.

Most of the compounds associated with microbiome presence, diversity, or composition were found in lower abundance in the plasma of HU mice, relative to LCM mice. For example, lactiflorin was nearly seven times less abundant in HU mice. Our data suggest that the presence of a gut microbiome either (i) inhibits the absorption of lactiflorin, (ii) increases metabolism of lactiflorin by the host, or (iii) allows for bacterial degradation of lactiflorin. In fact, most compounds associated with microbiome presence, diversity, or composition were less abundant in HU relative to LCM mice, suggesting potential co-metabolism of these compounds by the microbiome.

Another compound identified to be associated with microbiome composition was gamma-glutamyl-alanine. Notably, many plants and mammals possess gamma-glutamyl transpeptidase (GGT) (59), which is required for the synthesis of gamma-glutamyl amino

acids and often involved in glutathione metabolism; however, most bacteria also encode GGT (60), making the microbiome a candidate to alter concentrations of gamma-glutamyl-alanine either *via* direct metabolism or *via* an unknown effect on host metabolism. We did not detect gamma-glutamyl-alanine in the plasma of mice not gavaged with GT; hence, we consider this to be gamma-glutamyl-alanine above baseline concentrations. Seventy percent of the variation in plasma gamma-glutamyl-alanine was explainable by the ratio of *Alistipes* to *Butyricimonas*. Notably, human and murine gut colonizers within the genus *Butyricimonas* contain the gene gamma-D-glutamyl-L-lysine dipeptidyl-peptidase (K20742) (61), which releases gamma-glutamyl-alanine from cell walls, and *Alistipes* is negatively correlated with plasma GGT in humans (62), supporting the directionality of the relationship of this ratio with plasma gamma-glutamyl alanine abundance. However, this balance may just be a proxy for overall microbiome functional composition, as other members of the microbial community that interact with *Alistipes* and *Butyricimonas* may be more directly responsible for differences in gamma-glutamyl alanine.

We also found many significant relationships (after FDR correction) between GT compounds and individual taxa when only HU mice were included in the analysis (Fig. 8). In contrast to the multivariate associations where most compounds were lower in HU mice, we found primarily positive associations between individual taxa and GT compounds in plasma. The annotated GT compound with the highest number of significant relationships with individual taxa was spiramycin, an antimicrobial produced by *Streptomyces ambofaciens*, a member of the soil microbiome/rhizosphere (63). However, the mechanism of the positive association between spiramycin and ASVs within the genera *Gemmiger* and *Lactobacillus* is unclear. We found positive relationships with another bioactive plant compound, wilforlide B (a norditerpenoid) and *Anaerohabdus*. Although the data are compelling in that the majority of compounds have been found in plants, the mechanistic relationships between these compounds and individual taxa are unclear, especially given that metabolism pathways are poorly understood. Future work could track closely related compounds or apply machine learning-based approaches to identify potential metabolic pathways for these natural products and assess microbial genomes for genes responsible for these reactions.

One limitation of the study is the small number of replicate mice that were colonized with a specific human microbiome ($n = 2$) and that mice were colonized with only 10 different human microbiomes. Although significant relationships between specific bacteria and GT compounds were found, increasing the sample size would increase the power to identify relationships, especially in the permutation-based tests, where we needed to reduce our effective sample size to control for pseudoreplication. More samples would potentially allow for using a within-estimator (within each humanized microbiome source) to estimate the effects of small changes to individual taxa within a similar microbiome community structure subtype; this is akin to estimating the effects of increasing one species of bird within different terrestrial biomes like grassland and savannah. Another weakness is the lack of plasma and microbiome sampling before green tea gavage. We removed any compounds that were detectable in (HU and LCM) mice that did not receive a GT gavage; however, having samples from mice before and after gavage would also increase our ability to identify metabolism of GT compounds. Our focus on only GT compounds found in plasma does reduce the possibility of assessing changes in metabolic products of GT compounds.

Moreover, we chose the 2 h time point to maximize the GT compound signal in plasma, although multiple plasma samples after feeding would allow for a more detailed resolution of GT compound abundance trajectories based on microbiome composition, including time lags to assess causal directionality. Although the 2 h time point is when most GT compounds were detectable in plasma in a small pilot study, it is not necessarily when the microbiome may have the strongest effects. Stronger doses of GT or more variety in time points may alter the effects of the microbiome on GT compounds in plasma. It should be noted that the dose of GT, if scaling linearly, is comparable with 1–3

servings in an adult human, or if scaling allometrically, is equivalent to a small serving of GT in an adult human.

An additional limitation of the study is the presence of gut microbiomes, albeit of low diversity, in the LCM mouse group. Although it was assumed that all mice were exposed to the same contaminated low polyphenol mouse food, the contaminating taxa were able to dominate the fecal microbiome of the LCM mice, such that there were no mice free of microbial metabolism of GT compounds. Current studies in the laboratory utilize a 2-week feeding of GT to compare the effects of the microbiome on sub-chronic versus acute metabolism of GT.

One strength of our experimental design lies in the use of 10 different human microbiomes to enable the evaluation of relationships between individual microbial species and GT compounds. Although many humanized microbiome studies are pseudoreplicated (43), our use of multiple donor microbiomes while controlling for replication covers a broad variety of possible human-like microbiomes. Together, these results suggest that specific bacterial species may affect the metabolism of these bioactive compounds, thereby influencing their health effects.

Overall, this study represents a successful workflow for discovering relationships between food compounds and the composition of the gut microbiome, as well as individual gut bacterial genera. In addition, this methodology has allowed us to track GT compounds from the tea to the plasma. Although our study was limited in size and time points, we believe this methodology can and should be applied to study the effects of the gut microbiome on food metabolism. We identified multiple relationships between microbiome composition and GT compounds in plasma after GT consumption, supporting that bacterial taxa affect the absorption and metabolism of GT compounds, thereby possibly influencing their positive or negative health effects.

## ACKNOWLEDGMENTS

This project was funded through NIH/NIDDK R01DK113957 (Reisdorph, Campbell, and Krebs) and an ALSAM Therapeutic Innovation Grant through the Skaggs School of Pharmacy and Pharmaceutical Sciences to Dr. Reisdorph. Support for this project was also provided by the National Science Foundation-sponsored Interdisciplinary Quantitative Biology PhD program, the Integrated Data Science (Int dS) Graduate Training Fellowship, and the William J. Freytag Fellowship.

## AUTHOR AFFILIATIONS

[1]Department of Integrative Physiology, University of Colorado, Boulder, Colorado, USA

[2]Interdisciplinary Quantitative Biology, University of Colorado, Boulder, Colorado, USA

[3]Skaggs School of Pharmacy and Pharmaceutical Sciences, University of Colorado, Aurora, Colorado, USA

[4]Department of Biomedical Informatics, Anschutz Medical Campus, University of Colorado, Aurora, CO

[5]Division of Rheumatology, Department of Medicine, University of Colorado, Aurora, Colorado, USA

## AUTHOR ORCIDs

John D. Sterrett http://orcid.org/0000-0002-0931-7181
Kristine A. Kuhn https://orcid.org/0000-0002-6900-4195
Catherine A. Lozupone http://orcid.org/0000-0003-4786-7202
Nichole A. Reisdorph http://orcid.org/0000-0002-0425-8012

## FUNDING

| Funder | Grant(s) | Author(s) |
|---|---|---|
| National Science Foundation (NSF) | | John D. Sterrett |

| Funder | Grant(s) | Author(s) |
|---|---|---|
| HHS | NIH | National Institute of Diabetes and Digestive and Kidney Diseases (NIDDK) | R01DK113957 | Nichole A. Reisdorph |

## AUTHOR CONTRIBUTIONS

John D. Sterrett, Conceptualization, Formal analysis, Investigation, Methodology, Project administration, Software, Validation, Visualization, Writing – original draft, Writing – review and editing | Kevin D. Quinn, Data curation, Investigation, Methodology | Katrina A. Doenges, Data curation, Investigation, Methodology, Writing – review and editing | Nichole M. Nusbacher, Data curation, Investigation, Methodology, Writing – review and editing | Cassandra L. Levens, Data curation, Investigation, Methodology | Mike L. Armstrong, Data curation, Formal analysis, Investigation, Methodology, Project administration, Writing – review and editing | Richard M. Reisdorph, Conceptualization, Data curation, Investigation, Project administration, Writing – review and editing | Harry Smith, Formal analysis, Investigation | Laura M. Saba, Conceptualization, Investigation, Methodology | Kristine A. Kuhn, Conceptualization, Investigation, Methodology, Supervision, Writing – review and editing | Catherine A. Lozupone, Conceptualization, Methodology, Project administration, Resources, Supervision, Writing – original draft, Writing – review and editing | Nichole A. Reisdorph, Conceptualization, Funding acquisition, Investigation, Methodology, Project administration, Resources, Supervision, Writing – original draft, Writing – review and editing

## DATA AVAILABILITY

Data and code for reproducing these analyses can be accessed at https://github.com/sterrettJD/GT-micro-metabo. Raw 16S sequencing data is available in the European Nucleotide Archive using the accession IDs PRJEB77100 and ERP161582.

## ADDITIONAL FILES

The following material is available online.

### Supplemental Material

**Supplemental methods (Spectrum01799-24-S0001.docx).** Supplemental information for experimental methodology.
**Supplemental material (Spectrum01799-24-S0002.docx).** Tables S1 to S3.
**Supplemental data (Spectrum01799-24-S0003.csv).** All compounds identified as green tea-derived compounds detected in plasma.

### Open Peer Review

**PEER REVIEW HISTORY (review-history.pdf).** An accounting of the reviewer comments and feedback.

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
