## [Reviewer comments · Microbiology Spectrum]

Microbiology Spectrum

Appearance of green tea compounds in plasma following acute green tea consumption is modulated by the gut microbiome in mice

John Sterrett, Kevin Quinn, Katrina Doenges, Nichole Nusbacher, Cassandra Levens, Mike Armstrong, Richard Reisdorph, Harry Smith, Laura Saba, Kristine Kuhn, Catherine Lozupone, and Nichole Reisdorph

Corresponding Author(s): Nichole Reisdorph, University of Colorado Anschutz Medical Campus

Review Timeline:

Submission Date:	July 18, 2024
Editorial Decision:	August 9, 2024
Revision Received:	October 17, 2024
Editorial Decision:	October 30, 2024
Revision Received:	November 12, 2024
Accepted:	November 19, 2024

Editor: Jennifer Auchtung

Reviewer(s): The reviewers have opted to remain anonymous.

Transaction Report:

DOI: <https://doi.org/10.1128/spectrum.01799-24>

Re: Spectrum01799-24 (Appearance of green tea compounds in plasma following green tea consumption is modulated by the gut microbiome in mice)

Dear Dr. John D Sterrett:

Thank you for the privilege of reviewing your work. I have received three reviews that reflect differences in the reviewers' perspectives of the strengths and weaknesses of your work. I echo the concerns of reviewers 1 and 3 that in the absence of metabolomics from "GF" and HU mice without green tea treatment, it is not possible to decisively conclude that compounds originate from green tea rather than other sources. To address this significant concern, you would need to provide that data in a revised manuscript - either from your own control mice or from previous metabolomics analysis from mice that show that these compounds are not detected. If you are unable to provide this important, the manuscript would not be acceptable for publication.

In addition, as mentioned by reviewers 1 and 3, data that is not statistically significant should not be presented as supporting your conclusions of trends.

One final minor point to address in addition to those raised by reviewers - as mice are not germ-free, do not refer to them as germ free or GF mice in the text. The presence of a simple community does not negate their utility,

Below you will find instructions from the Spectrum editorial office and the reviewer comments.

Revision Guidelines

Sincerely,
Jennifer Auchtung
Editor
Microbiology Spectrum

Reviewer #1 (Comments for the Author):

In the manuscript "Appearance of green tea compounds in plasma following green tea consumption is modulated by the gut microbiome in mice". The authors explored the relationships between gut microbiome and green compounds in the plasma for germ-free or human microbiome colonized mice. The study design is lack of rigor, and the conclusions are not supported by the results and/or the study design.

1. The study aimed to explore how gut microbiome modulate green tea consumption in mice. However, the hypothesis cannot be tested by the study design. The correlations between microbiome and plasma green tea compound level cannot indicate modulation relationship. In fact, Figure 3C, 3D shows no differences in green tea compounds between HC and GF mice, who harbor very distinct gut microbiome.
2. The study design is lack of rigor in that the plasma green tea compounds levels should be compared to that from pre-green tea gavaged mice, which is the baseline level.
3. Please remove results/statements that are not statistically significant and/or not significant by FDR p values.
4. In Figure 3D, there is clear separation of data on PCo2, what caused the two clusters of data?

Reviewer #2 (Comments for the Author):

Sterrett et al. explore the relationships between green tea (GT) compounds present in mice plasma due differences in gut microbiomes. The study used microbiome and metabolomics in low-diversity, pseudo-germ-free mice (GF) and human colonized (HU) gnotobiotic mice treated with an acute dose of matcha GT. The study is well designed to test their hypotheses, and the results are presented and discussed in a clear way. This reviewer suggests the following minor revisions:

Title (L1): Add "acute green tea consumption" to avoid misleading readers.

Abstract

L22: Change second "due to" in the sentence to "given" or similar to improve readability.

L42: Remove "often considered, such as those" from sentence to improve readability.

Introduction

L62-65: Move "Our approach addresses this challenge through use of human colonized (HU) mice fed a food with known health effects, namely green tea (GT). It aims to determine how the gut microbiome affects which GT compounds are found in the plasma of HU and germ free (GF) mice." to the last paragraph so that readers can understand the background before getting to your specific work.

L76: check reference 21. In the previous line Wu et al is followed by reference 20.

L90-92: Modify sentence to describe the study objectives without mentioning details that are described in the methods section.

L92-95. This information is unnecessary in the introduction since it is described in methods.

Methods:

L101, L113, L134: Check for consistent use of Oxford commas.

L101-109. You indicate the vendors of the chemical reagents but not the brands you specifically use. Please modify to show the brands of the chemicals and reagents used to increase reproducibility of the study.

L132: Indicate the brand of the green tea used to increase reproducibility of the study.

L150: Where are the four GT extracts used in metabolomics coming from? Are this coming from each mice cohort group plus initial GT product? Please clarify.

Results:

L312: "Figure 1". This figure shows a schematic of the methods, but not of the results described. I suggest you move this figure to the methods sections and add a supplemental table or figure showing the identity of the 624 GT compounds that were further evaluated in the study.

L341: What do you mean by "very similar"? Do you have any statistical results to support this observation?

L343: Please detail what you mean by "high relative abundances". What percentages are considered as high in this study?

L368: "stronger relationship". Please clarify that this relationship is not significant based on your statistical results.

L406: "strongest associations". Do you have statistics showing the strength of the association? Please add to aid the reader in interpreting your results.

Discussion:

L503: "70%". Spell out numbers at the beginning of a sentence.

L534-558: In the limitations paragraph, I believe you should discuss the dose of GT extract used in the study. How does it compare to a normal intaking of GT in humans, and would you expect the same responses at lower doses?

Figure 6, 8, and 9. The font in the figure is too small. Please modify so readers can read the axis and correctly interpret your results.

Supplemental Figure 1: Figure is empty - I can only see the outline of the plot, but no samples in the PCA.

Reviewer #3 (Comments for the Author):

Sterrett et al. explored the relationship between the gut microbiome and green tea compounds in plasma using gnotobiotic mice colonized with human gut microflora. They measured plasma metabolites and fecal bacteria two hours after green tea administration and employed linear regressions and mixed models to identify microbial taxa correlated with the abundances of green tea compounds in plasma. While the large-scale microbiome-metabolite analysis offers valuable insights into the interactions between food compounds and the gut microbiome, there are some concerns regarding the study's reliability due to limited statistical power and the absence of certain controls.

1. Although germ-free mice with green tea administration were used as controls, an additional control group of humanized mice without green tea treatment would have been critical. This could have better established the effects of green tea on the gut microbiome and confirmed that the metabolites detected in plasma were truly derived from green tea.
2. Many analyses yielded non-significant results after multiple comparison correction (Figs. 3,4,5), reflecting the study's limited statistical power. Additionally, it remains uncertain whether the two-hour time point is sufficient to detect significant changes in the fecal microbiome, which could explain the lack of microbiome-associated metabolites observed.
3. Faith's PD is a diversity metric sensitive to sequencing depth. To strengthen the data's robustness, the authors are encouraged to provide sequencing depth for each sample or consider using alternative diversity metrics, such as the Shannon or Simpson index, which are less affected by sequencing depth.

Dear Dr. Auchtung,

Thank you for the opportunity to revise this work. We believe this review process has improved the quality of the research presented, so we are grateful for the opportunity and hope that our revisions sufficiently meet the expectations of you and the reviewers. We have included a point-by-point response to your comments and those of the reviewers below.

Editor comment:

I echo the concerns of reviewers 1 and 3 that in the absence of metabolomics from "GF" and HU mice without green tea treatment, it is not possible to decisively conclude that compounds originate from green tea rather than other sources. To address this significant concern, you would need to provide that data in a revised manuscript - either from your own control mice or from previous metabolomics analysis from mice that show that these compounds are not detected. If you are unable to provide this important, the manuscript would not be acceptable for publication.

Response:

We agree that this was a major limitation of the originally submitted manuscript. We have addressed this by removing any compounds that were found in the plasma of comparable mice who were not gavaged with green tea in all presented data analyses, and the results of the paper have been modified accordingly. we removed 86 lipid compounds and 59 aqueous compounds from the original 432 lipid compounds and 192 aqueous compounds. We believe that this has greatly improved the rigor of our study, and the overall conclusions of the paper have not substantially changed. A description of the mice used for assessing metabolome in the absence of green-tea gavage, and our methods for identifying endogenous compounds, have been added to the methods and supplemental methods sections of the manuscript.

The following was added to the Supplemental Methods and an abbreviated version was added to the Methods section. (Note: we have updated "germ free" to "low complexity microbiome", abbreviated LCM, in response to feedback on the use of "germ free".)

Preliminary experiment using mice not fed GT: A preliminary experiment analyzing plasma metabolomes from LCM and HU mice not fed green tea was conducted to determine baseline differences. As with the main study, mice were placed on an irradiated, low polyphenol diet (TD.97184, Envigo, Indianapolis, IN) at the time of weaning (3 weeks of age) through 6 weeks of age. At 4 weeks of age, mice were orally gavaged with 200 μ L of either human fecal slurry (100 mg stool homogenized in 1 mL reduced PBS in an anaerobic chamber) or PBS for LCM. For this experiment, a total of 16 mice were used; 6 mice were gavaged with vehicle and 10 mice were gavaged with the same 10 distinct human fecal slurries as described in the main text (i.e. 1 microbiome per mouse). At two weeks post-gavage, plasma was collected as described

in the main text. Metabolomics of plasma was performed as described in the main text and above using LC/MS. Database searching was performed in ID Browser (Agilent Technologies) as described in the main text. Briefly, match criteria included mass (<10 ppm window), isotope peak abundance and spacing, and a match score cutoff of 60. Compounds were first searched against in-house standards databases, containing over 600 compounds. For in-house standards databases retention time was required for a match with a tolerance window of 0.3 minutes. Unannotated compounds were then searched against FooDB, METLIN, Human Metabolome Database (HMDB), Kyoto Encyclopedia of Genes and Genomes (KEGG), and Lipid Maps. For compounds matched to more than one database, the hit with the highest score was assigned as the annotation. Compounds that matched to a database entry represent a Metabolomics Standards Initiative level 3 identification, based on the proposed minimum reporting by Sumner, et al. and hence compound names are considered tentative.

Matching compounds from the GT mice and the non-GT controls: For each compound found in the plasma of mice fed GT (n=624 compounds), the monoisotopic ion was used to mine the plasma metabolomics data from the GF and HU mice not fed GT. Briefly, extracted ion chromatograms are manually generated for each green tea compound using non-GT plasma LC/MS data employing a targeted data extraction strategy using Qualitative Analysis (Agilent). For each GT compound extracted ion chromatogram, the MS spectrum was manually inspected to verify the targeted ion was present as a monoisotopic peak. Only quantifier/qualifier peaks detected within +/- 0.2 minutes of the retention time recorded in samples are retained. Briefly, if the monoisotopic ion for a GT compound was present in the non-GT data (as a monoisotopic peak only) and the retention time was within 0.2 min, it was considered a match. In all cases the mass error was less than 5 ppm.

Editor comment:

In addition, as mentioned by reviewers 1 and 3, data that is not statistically significant should not be presented as supporting your conclusions of trends.

Response:

We have updated the manuscript accordingly.

Largely, this includes rephrasing any sections of the results (particularly the subsection *Diversity, composition and relationships between the fecal microbiome, plasma metabolome, and plasma lipidome* and the subsection *Green tea compounds associated with microbiome composition*) to specify that results are not significant.

In accordance with this, we have removed the previous figure 5, which showed insignificant results and did not greatly contribute to the article. Supplemental table 2 still is included as a tabular depiction of the data shown in the original figure 5.

Additionally, we have removed sections of the discussion that were previously discussing insignificant data.

We believe that this has strengthened the paper in focusing on significant results that our study was powered to detect, so we appreciate this comment from you and the reviewers.

Editor comment:

One final minor point to address in addition to those raised by reviewers - as mice are not germ-free, do not refer to them as germ free or GF mice in the text. The presence of a simple community does not negate their utility.

Response:

We agree that the use of “germ-free” may lead to confusion or misinterpreted results. Throughout the paper we have now renamed these mice as “low complexity microbiome mice”, abbreviated as “LCM”.

Reviewer #1 comment:

In the manuscript "Appearance of green tea compounds in plasma following green tea consumption is modulated by the gut microbiome in mice". The authors explored the relationships between gut microbiome and green compounds in the plasma for germ-free or human microbiome colonized mice. The study design is lack of rigor, and the conclusions are not supported by the results and/or the study design.

1. The study aimed to explore how gut microbiome modulate green tea consumption in mice. However, the hypothesis cannot be tested by the study design. The correlations between microbiome and plasma green tea compound level cannot indicate modulation relationship. In fact, Figure 3C, 3D shows no differences in green tea compounds between HC and GF mice, who harbor very distinct gut microbiome.

Response:

The reviewer states, “the study aimed to explore how gut microbiome modulate green tea consumption in mice.” However, that was not the goal of our study, which was to study how the gut microbiome affects the concentrations of green tea-derived compounds in plasma following standardized consumption of green tea.

We do believe that this study was poised to study this question, as all variables were held constant across the experimental mice, other than the human microbiomes that they were colonized with.

Indeed, we did not change the existing microbiome within each of the experimental mice to assess within a mouse how alterations to the microbiome affect green tea

compounds in plasma. However, gnotobiotic mice are a suitable option for assessing the effects of various microbiome compositions on biological processes, as non-microbial confounders are held constant.

Moreover, looking at only Figure 3C/D does not mean that there are no associations between plasma green tea compounds and the gut microbiome, as those PCoA plots show the multivariate similarity between samples. It is very likely that many compounds would be unaffected by microbiome composition, as many food-derived compounds are absorbed in the small intestine. Further, this was an acute dosing study, whereby samples were collected 2 hours post-gavage. That is the reason that only a small portion of the paper focuses on associations between the multivariate composition of the green tea compounds in plasma with the microbiome, and instead we analyzed single compounds at a time.

2. The study design is lack of rigor in that the plasma green tea compounds levels should be compared to that from pre-green tea gavaged mice, which is the baseline level.

Response:

We agree that there is a weakness of the paper, in that compounds that could have come from endogenous, environmental, and dietary sources were included in analyses. This weakness was discussed in the previous draft and thoroughly acknowledged.

However, in this revised draft, we have been able to mitigate this weakness using metabolomics data that we have collected from mice not gavaged with green tea (which would presumably be endogenous to the mice, their environments, or their diets). Specifically, we have removed from consideration all compounds that were found in these mice. This has resulted in the removal of 86 lipid compounds and 59 aqueous compounds from the original 432 lipid compounds and 192 aqueous compounds. This is described in the methods (subsection *Green Tea Compounds in Plasma*), Supplemental Methods, and results (subsection *Metabolomics of mouse plasma and GT*).

3. Please remove results/statements that are not statistically significant and/or not significant by FDR p values.

Response:

We have updated the manuscript accordingly.

From our response to the editor's comments:

Largely, this includes rephrasing any sections of the results (particularly the subsection *Diversity, composition and relationships between the fecal microbiome, plasma metabolome, and plasma lipidome* and the subsection *Green tea compounds associated with microbiome composition*) to specify that results are not significant.

In accordance with this, we have removed the previous figure 5, which showed insignificant results and did not greatly contribute to the article. Supplemental table 2 still is included as a tabular depiction of the data shown in the original figure 5.

Additionally, we have removed sections of the discussion that were previously discussing insignificant data.

We believe that this has strengthened the paper in focusing on significant results that our study was powered to detect, so we appreciate this comment.

4. In Figure 3D, there is clear separation of data on P_{CO2}, what caused the two clusters of data?

Response:

In an attempt to explain this clustering, we assessed this separation based on the sample metadata available, such as experimental cohort, and did not find that any columns predicted this separation.

This is described in the Supplemental Methods, subsection Effects of cohort on composition of the fecal microbiome, plasma GT metabolome, and plasma GT lipidome.

Reviewer #2 comment:

Sterrett et al. explore the relationships between green tea (GT) compounds present in mice plasma due differences in gut microbiomes. The study used microbiome and metabolomics in low-diversity, pseudo-germ-free mice (GF) and human colonized (HU) gnotobiotic mice treated with an acute dose of matcha GT. The study is well designed to test their hypotheses, and the results are presented and discussed in a clear way. This reviewer suggests the following minor revisions:

Reviewer #2 comment:

Title (L1): Add "acute green tea consumption" to avoid misleading readers.

Response:

This has been updated in accordance with the reviewer's suggestion.

Reviewer #2 comment:

Abstract

L22: Change second "due to" in the sentence to "given" or similar to improve readability.

Response:

This has been updated in accordance with the reviewer's suggestion.

Reviewer #2 comment:

L42: Remove "often considered, such as those" from sentence to improve readability.

Response:

This has been updated in accordance with the reviewer's suggestion.

Reviewer #2 comment:

L62-65: Move "Our approach addresses this challenge through use of human colonized (HU) mice fed a food with known health effects, namely green tea (GT). It aims to determine how the gut microbiome affects which GT compounds are found in the plasma of HU and germ free (GF) mice." to the last paragraph so that readers can understand the background before getting to your specific work.

L90-92: Modify sentence to describe the study objectives without mentioning details that are described in the methods section.

L92-95. This information is unnecessary in the introduction since it is described in methods.

Response:

We have considered these three helpful points together. The introduction section has been altered to include fewer details about our study, and details on our methods have been removed from the introduction. Additionally, statements regarding the scope of the project have been moved to the last paragraph of the introduction, as suggested.

Reviewer #2 comment:

L76: check reference 21. In the previous line Wu et al is followed by reference 20.

Response:

Thank you, we have clarified the relevance of citation 21 in the introduction section of the paper.

Reviewer #2 comment:

Methods:

L101, L113, L134: Check for consistent use of Oxford commas.

Response:

Thank you for catching this discrepancy. We have updated the manuscript to be consistent in our use of the Oxford comma.

Reviewer #2 comment:

L101-109. You indicate the vendors of the chemical reagents but not the brands you specifically use. Please modify to show the brands of the chemicals and reagents used to increase reproducibility of the study.

Response:

We appreciate this comment and help in ensuring the reproducibility of the study. In our descriptions, in many cases the brands provided here are vendor-specific brands. For example, the Burdick & Jackson water is from Burdick & Jackson. In cases where the brands are available, we have provided them. The Methods subsection Chemicals, standards, and reagents now reads as follows:

Chemicals, standards, and reagents used for sample preparation and liquid chromatography-mass spectrometry (LCMS) analysis were of high-performance liquid chromatography (HPLC) or LCMS grade. These included water from Honeywell Burdick & Jackson (Muskegon, MI, USA), J.T. Baker methyl tert-butyl ether (MTBE) from VWR (Radnor, PA, USA), formic acid from ThermoFisher Scientific (Waltham, MA, USA),

Fisher Chemical acetonitrile and methanol from Fisher Scientific (Fair Lawn, NJ, USA), and Supelco 2-Propanol from Millipore Sigma (Burlington, MA, USA). Authentic standards for sample preparation were from Avanti Polar Lipids Inc. (Alabaster, AL, USA), Cambridge Isotope Laboratories (Tewksbury, MA, USA), Sigma-Aldrich (St. Louis, MO, USA), and CDN Isotopes (Pointe-Claire, Quebec, Canada).

Reviewer #2 comment:

L132: Indicate the brand of the green tea used to increase reproducibility of the study.

Response:

The green tea was Jade Leaf Matcha Organic Green Tea Powder - Culinary Grade Premium Second Harvest - Authentic Japanese Origin (8.8 Ounce Pouch), purchased from Amazon.com, to be comparable to any likely human-consumed green tea. We have clarified this in the methods section.

Reviewer #2 comment:

L150: Where are the four GT extracts used in metabolomics coming from? Are this coming from each mice cohort group plus initial GT product? Please clarify.

Response:

The four GT extracts used in metabolomics were prepared in the same batch as the other GT gavages and analyzed in one batch, rather than being spread across the cohorts. This has been clarified in the methods, in the last sentence of the *Mouse GT Gavage* section, which now states "Four leftover slurries were stored at -80 °C for GT metabolomics analysis."

Reviewer #2 comment:

Results:

L312: "Figure 1". This figure shows a schematic of the methods, but not of the results described. I suggest you move this figure to the methods sections and add a supplemental table or figure showing the identity of the 624 GT compounds that were further evaluated in the study.

Response:

We agree that an overview of the methods is shown here. However, we believe that referencing this figure at the end of the introduction section provides a graphical depiction of the methods for the reader, which serves as a primer before the methods section. We are open to moving the reference to this figure to the methods but would prefer to leave it as it currently as, as we believe it improves clarity of the article.

We have provided a table of compound names as a separate supplemental file. It can also be accessed per the data and code availability statement.

Reviewer #2 comment:

L341: What do you mean by "very similar"? Do you have any statistical results to support this observation?

Response:

We did not originally quantify "very similar", but we now have quantified this using a PERMANOVA, which identified that the variation between donor ID pairs is much larger than the variation within donor ID pairs. We have also added quantification of the UniFrac distance within versus between donor IDs. That sentence has been changed to say:

"Microbiome composition was very similar within HU mouse replicates (UniFrac distance within pairs mean [95% confidence interval (CI)] = 0.25 [0.21 – 0.30]), and variable between mice colonized with different human donors (UniFrac distances between centroids mean [95% CI] = 0.50 [0.49 – 0.52]) (Figure 2). This is supported by a PERMANOVA demonstrating larger variance in phylogenetic composition between sample pairs than within pairs ($R^2 = 0.87$, pseudo- $F = 7.58$, $p < 0.001$)."

Reviewer #2 comment:

L343: Please detail what you mean by "high relative abundances". What percentages are considered as high in this study?

Response:

The phrase high relative abundance is used twice in this section, and though the comment is only referring to one of these instances, we have addressed both.

We have updated "Additionally, LCM mice had high relative abundances of *Epulopiscium* (ANCOM-BC; $p = 0.015$), and *Turicibacter* (ANCOM-BC; $p = 0.002$) in their microbiomes compared to HU mice (Figure 2)." to say "higher" instead of "high".

Regarding "The microbiome of all but two pairs of HU mice had *Akkermansia* present at high relative abundances."

We considered $\geq 20\%$ relative abundance of *Akkermansia* to be high. This is visible in the taxonomy bar plot (Figure 2), but we have also specified it in that sentence, which now reads, "The microbiome of all but two pairs of HU mice had *Akkermansia* present at high relative abundances ($\geq 20\%$)."

Reviewer #2 comment:

L368: "stronger relationship". Please clarify that this relationship is not significant based on your statistical results.

Response:

This has been updated to say "Furthermore, a Procrustes randomization test did not reveal an overall significant relationship between microbiome composition (UniFrac) and GT lipids in plasma (**Figure 4A**, $m^2 = 0.39$, $p = 0.085$) or aqueous GT compounds in plasma (using Procrustes-transformed Bray-Curtis PCoA, **Figure 4B**, $m^2 = 0.66$, $p = 0.79$)."

Reviewer #2 comment:

L406: "strongest associations". Do you have statistics showing the strength of the association? Please add to aid the reader in interpreting your results.

Response:

We appreciate this comment to improve readability of this article. Though this information is available in the supplemental material, we have added descriptors of statistical associations to the body of the results.

Reviewer #2 comment:

Discussion:

L503: "70%". Spell out numbers at the beginning of a sentence.

Response:

We thank the reviewer for this comment. This sentence has been updated accordingly

Reviewer #2 comment:

L534-558: In the limitations paragraph, I believe you should discuss the dose of GT extract used in the study. How does it compare to a normal intaking of GT in humans, and would you expect the same responses at lower doses?

Response:

This is a great point that is very useful in the discussion. We used a green tea extract that was prepared based on 50 mg/kg of mouse body weight. This value was obtained from a pilot study that utilized 0, 5, 20, or 50 mg/kg green tea (Supplemental Methods) and supported by literature.^{1,2}

Scaling linearly, this is equivalent to a 3g dose of matcha in a 60kg human, which would constitute 2-3 servings of matcha.

However, allometric scaling identifies this as a lower dose in humans. Using the following formula³ results in a human equivalent dose (HED) of 3.5 mg/kg.

$$\text{HED} = (\text{mg/kg} = \text{Animal dose mg/kg}) \times (\text{Weight}_{\text{animal}} [\text{kg}]/\text{Weight}_{\text{human}} [\text{kg}])^{(1-0.67)}$$

In a 60 kg human, this would be a small dose of matcha of 200 mg, which would constitute less than one serving.

Unfortunately, no standards have been established on how microbiome-relevant dosing scales between species, and some of the motivations for allometric scaling (differences in metabolic rate) may not apply to the microbiome, for which the metabolic rate does not necessarily scale relative the mass:surface area ratio. Additionally, not all drug/food dosing should scale allometrically, and sometimes linear scaling provides a more accurate dose scaling.⁴

We have added a brief discussion of this to the discussion section, which reads: “Stronger doses of GT or more variety in timepoints may have alter the effects of the microbiome on GT compounds in plasma. It should be noted that the dose of GT, if scaling linearly, is comparable to 1-3 servings in an adult human, or if scaling allometrically, is equivalent to a small serving of GT in an adult human.”

1. Di Lorenzo A, Nabavi SF, Sureda A, Moghaddam AH, Khanjani S, Arcidiaco P, Nabavi SM, Daglia M. Antidepressive-like effects and antioxidant activity of green tea and GABA green tea in a mouse model of post-stroke depression. *Mol Nutr Food Res*. 2016 Mar;60(3):566-79. doi: 10.1002/mnfr.201500567. Epub 2015 Dec 29. PMID: 26626862.
2. Zhu WL, Shi HS, Wei YM, Wang SJ, Sun CY, Ding ZB, Lu L. Green tea polyphenols produce antidepressant-like effects in adult mice. *Pharmacol Res*. 2012 Jan;65(1):74-80. doi: 10.1016/j.phrs.2011.09.007. Epub 2011 Sep 22. PMID: 21964320.
3. Nair AB, Jacob S. A simple practice guide for dose conversion between animals and human. *J Basic Clin Pharm*. 2016 Mar;7(2):27-31. doi: 10.4103/0976-0105.177703. PMID: 27057123; PMCID: PMC4804402.
4. Sharma V, McNeill JH. To scale or not to scale: the principles of dose extrapolation. *Br J Pharmacol*. 2009 Jul;157(6):907-21. doi: 10.1111/j.1476-5381.2009.00267.x. Epub 2009 Jun 5. PMID: 19508398; PMCID: PMC2737649.

Reviewer #2 comment:

Figure 6, 8, and 9. The font in the figure is too small. Please modify so readers can read the axis and correctly interpret your results.

Response:

We have modified these figures (now figures 5, 7, and 8) accordingly. With the new results, we are displaying less data (due to fewer significant relationships), which makes the possible font size larger, and we have increased it where possible. Unfortunately, the amount of information that needs to fit on these figures makes it difficult to increase all font sizes without occluding the data, so we haven't been able to increase the size of taxonomic labels on Figure 6, but we have increased other text sizes.

Reviewer #2 comment:

Supplemental Figure 1: Figure is empty - I can only see the outline of the plot, but no samples in the PCA.

Response:

Our plot does have data shown. We believe there must have been an issue with the file that reviewer #2 downloaded, or an issue with any file previewing, but when we re-download the files from the portal, Figure 1 was not empty. Below is a screenshot of what was in the Supplemental Methods file downloaded from the author portal.

Supplemental Figure 1: Using whole metabolome data (rather than green tea compounds only), PCA of

This figure has been updated to use the terminology “low complexity” to refer to the microbiomes of the mice that were previously referred to as germ-free. Another

screenshot of the resubmitted file can be seen below.

Supplemental Figure 1: Using whole metabolome data (rather than green tea compounds only), PCA of

Reviewer #3 comment:

Sterrett et al. explored the relationship between the gut microbiome and green tea compounds in plasma using gnotobiotic mice colonized with human gut microflora. They measured plasma metabolites and fecal bacteria two hours after green tea administration and employed linear regressions and mixed models to identify microbial taxa correlated with the abundances of green tea compounds in plasma. While the large-scale microbiome-metabolite analysis offers valuable insights into the interactions between food compounds and the gut microbiome, there are some concerns regarding the study's reliability due to limited statistical power and the absence of certain controls.

Reviewer #3 comment:

1. Although germ-free mice with green tea administration were used as controls, an additional control group of humanized mice without green tea treatment would have been critical. This could have better established the effects of green tea on the gut microbiome and confirmed that the metabolites detected in plasma were truly derived from green tea.

Response:

We agree with this comment and appreciate it being highlighted by this reviewer. To address this weakness, we have now removed from consideration all compounds that were found in mice not gavaged with green tea (which would presumably be endogenous to the mice, their environments, or their diets). This has resulted in the removal of 86 lipid compounds and 59 aqueous compounds from the original 432 lipid compounds and 192 aqueous compounds. This is described in the methods (subsection *Green Tea Compounds in Plasma*), Supplemental Methods and results (subsection *Metabolomics of mouse plasma and GT*).

We did not see large changes in the results of multivariate analyses. However, some of the previously identified relationships between GT and the microbiome have been removed from consideration because those compounds were found in mice not gavaged with GT.

Reviewer #3 comment:

2. Many analyses yielded non-significant results after multiple comparison correction (Figs. 3,4,5), reflecting the study's limited statistical power. Additionally, it remains uncertain whether the two-hour time point is sufficient to detect significant changes in the fecal microbiome, which could explain the lack of microbiome-associated metabolites observed.

Response:

We have removed from the text discussion of “trending” results and have increased the focus on the results that were significant after multiple test correction.

From our response to the editor’s comment:

Largely, this includes rephrasing any sections of the results (particularly the subsection *Diversity, composition and relationships between the fecal microbiome, plasma metabolome, and plasma lipidome* and the subsection *Green tea compounds associated with microbiome composition*) to specify that results are not significant.

In accordance with this, we have removed the previous figure 5, which showed insignificant results and did not greatly contribute to the article. Supplemental table 2 still is included as a tabular depiction of the data shown in the original figure 5.

Additionally, we have removed sections of the discussion that were previously discussing insignificant data.

We believe that this has strengthened the paper in focusing on significant results that our study was powered to detect, so we appreciate this comment.

Moreover, we appreciate this reviewer's point about the 2 hour timepoint. As detailed in the Supplemental Methods, due to the prohibitive costs of humanized mouse studies, a pilot study was conducted in wild type mice that resulted in more detectable green tea compounds in plasma than the 4 or 24 hour timepoints. This point is also discussed in the manuscript (methods, subsection *Plasma and Tissue Collection*). However, we agree this is not a comprehensive assessment of all possible timepoints, nor did we assess which timepoint had the largest effect from the microbiome. A more extensive discussion of this has been added in the discussion section of the paper, which now reads:

"Additionally, while the two-hour timepoint is when the most GT compounds were detectable in plasma, it is not necessarily when the microbiome may have the strongest effects. Stronger doses of GT or more variety in timepoints may alter the effects of the microbiome on GT compounds in plasma."

Reviewer #3 comment:

3. Faith's PD is a diversity metric sensitive to sequencing depth. To strengthen the data's robustness, the authors are encouraged to provide sequencing depth for each sample or consider using alternative diversity metrics, such as the Shannon or Simpson index, which are less affected by sequencing depth.

Response:

It is true that Faith's PD is sensitive to sequencing depth. All diversity metrics used in this manuscript were subsampled (rarefied) to the same sequencing depth, so there isn't bias due to differences in sequencing depth between samples. This is described in the methods, subsection *Microbiome - Sequence data processing*, which states that "For diversity analyses, samples were rarefied to 40,625 reads per sample, which was the highest rarefaction depth that did not exclude any samples."

Re: Spectrum01799-24R1 (Appearance of green tea compounds in plasma following acute green tea consumption is modulated by the gut microbiome in mice)

Dear Dr. John D Sterrett:

Thank you for the privilege of reviewing your work. Below you will find some final proposed revisions from the reviewers along with instructions from the Spectrum editorial office.

Revision Guidelines

Sincerely,
Jennifer Auchtung
Editor
Microbiology Spectrum

Reviewer #1 (Comments for the Author):

All the comments were addressed and the manuscript was revised accordingly.

Reviewer #2 (Comments for the Author):

The authors addressed all the concerns presented by previous reviewers and adequately presented the limitations of the study. I have some minor comments:

L31-33: "We found multiple green tea compounds 32 in plasma associated with microbiome presence and diversity; we also detected strong 33 associations between green tea compounds in plasma and specific taxa in the gut". This line is too general for the abstract. Provide main findings of the study, pointing out which compounds associated with specific taxa.

L462-466: ". Despite using phylogenetic composition of the microbiome in our analyses, we did not see multivariate relationships between green tea compounds in plasma and the composition of the gut microbiome, which could be due to the absorption of many green tea compounds in the small intestine, before reaching the majority of microbial biomass in the colon". It is unclear what the authors mean by "multivariate relationships". Where you trying to identify correlations between the whole microbiota and the whole metabolome? Please clarify.

Figure 3 still has "germ-free" labels in panels A, B, and C

Figure 5: The font is too small to read. Consider making a 3x4 distribution of the panels to be able to increase the font size.

We appreciate the feedback from the reviewers. Comments from reviewer #2 have been addressed in the resubmitted manuscript.

Reviewer #1 (Comments for the Author):

All the comments were addressed and the manuscript was revised accordingly.

Response: We appreciate this reviewer's time spent improving the quality of our research. Thank you.

Reviewer #2 (Comments for the Author):

Reviewer comment: The authors addressed all the concerns presented by previous reviewers and adequately presented the limitations of the study.

Response: We also appreciate this reviewer's time spent improving the quality of our research. The further comments have continued to be helpful for communicating our work effectively. Thank you.

Reviewer comment: L31-33: "We found multiple green tea compounds 32 in plasma associated with microbiome presence and diversity; we also detected strong 33 associations between green tea compounds in plasma and specific taxa in the gut". This line is too general for the abstract. Provide main findings of the study, pointing out which compounds associated with specific taxa.

Response: This is helpful feedback. We have updated this to say the following: "We found multiple green tea compounds in plasma associated with microbiome presence and diversity (including acetylcholine, lactiflorin, and aspartic acid negatively associated with diversity). Additionally, we detected strong associations between bioactive green tea compounds in plasma and specific gut bacteria, including associations between spiramycin and *Gemmiger*, and between wildforlide and *Anaerorhabdus*."

Reviewer comment: L462-466: ". Despite using phylogenetic composition of the microbiome in our analyses, we did not see multivariate relationships between green tea compounds in plasma and the composition of the gut microbiome, which could be due to the absorption of many green tea compounds in the small intestine, before reaching the majority of microbial biomass in the colon". It is unclear what the authors mean by "multivariate relationships". Were you trying to identify correlations between the whole microbiota and the whole metabolome? Please clarify.

Response: We have clarified this in the text, which now reads as follows: "Despite using phylogenetic composition of the microbiome in our analyses, we did not

see multivariate-multivariate relationships between overall composition of green tea compounds in plasma and the composition of the gut microbiome (per a Procrustes randomization test on PCoA coordinates from green tea compound Bray-Curtis distance and Unweighted UniFrac microbiome distance), which could be due to the absorption of many green tea compounds in the small intestine, before reaching the majority of microbial biomass in the colon.”

Reviewer comment: Figure 3 still has "germ-free" labels in panels A, B, and C

Response: We have been unable to see where “germ-free” can be found in this figure. We’ve carefully searched the manuscript for uses of “germ-free” and cannot find any incorrect uses. If we have missed something, could you please point out exact where? We have included a screenshot of Figure 3 from the submitted manuscript below.

Reviewer comment: Figure 5: The font is too small to read. Consider making a 3x4 distribution of the panels to be able to increase the font size.

Response: We have now altered this figure to 2x6 instead of 4x3 to increase font size. Thank you for this suggestion. A low-resolution version of this figure is included below (higher resolution is included in the manuscript).

Re: Spectrum01799-24R2 (Appearance of green tea compounds in plasma following acute green tea consumption is modulated by the gut microbiome in mice)

Dear Dr. Nichole Reisdorph:

Your manuscript has been accepted, and I am forwarding it to the ASM production staff for publication. Your paper will first be checked to make sure all elements meet the technical requirements. ASM staff will contact you if anything needs to be revised before copyediting and production can begin. Otherwise, you will be notified when your proofs are ready to be viewed.

Sincerely,
Jennifer Auchtung
Editor
Microbiology Spectrum